# A Review on Inorganic Nanoparticles Modified Composite Membranes for Lithium-Ion Batteries: Recent Progress and Prospects

**DOI:** 10.3390/membranes9070078

**Published:** 2019-07-02

**Authors:** Muhammad Rehman Asghar, Muhammad Tuoqeer Anwar, Ahmad Naveed

**Affiliations:** 1Institute of Fuel Cells, School of Mechanical Engineering, Shanghai Jiao Tong University, MOE Key Laboratory of Power Machinery and Engineering, Shanghai Jiao Tong University, Shanghai 200240, China; 2COMSATS University Islamabad (Sahiwal Campus), off G.T Rd., Sahiwal, Punjab 57000, Pakistan; 3Shanghai Electrochemical Energy Devices Research Center, School of Chemistry and Chemical Engineering, Shanghai Jiao Tong University, Shanghai 200240, China

**Keywords:** lithium-ion battery, composite membranes, inorganic nanoparticles

## Abstract

Separators with high porosity, mechanical robustness, high ion conductivity, thin structure, excellent thermal stability, high electrolyte uptake and high retention capacity is today’s burning research topic. These characteristics are not easily achieved by using single polymer separators. Inorganic nanoparticle use is one of the efforts to achieve these attributes and it has taken its place in recent research. The inorganic nanoparticles not only improve the physical characteristics of the separator but also keep it from dendrite problems, which enhance its shelf life. In this article, use of inorganic particles for lithium-ion battery membrane modification is discussed in detail and composite membranes with three main types including inorganic particle-coated composite membranes, inorganic particle-filled composite membranes and inorganic particle-filled non-woven mates are described. The possible advantages of inorganic particles application on membrane morphology, different techniques and modification methods for improving particle performance in the composite membrane, future prospects and better applications of ceramic nanoparticles and improvements in these composite membranes are also highlighted. In short, the contents of this review provide a fruitful source for further study and the development of new lithium-ion battery membranes with improved mechanical stability, chemical inertness and better electrochemical properties.

## 1. Introduction

The development of sustainable and new kinds of energy production technology is a critical issue and goal of today’s world. Electric vehicles, as well as hybrid electric vehicles and electronic devices, are running on lithium-ion batteries, which are a very promising and efficient technology and the demand is increasing day by day in the market [1] since Sony Energy research scientists manufactured the first commercial lithium batteries in the late 1980s [2,3,4]. The use of the Lithium-ion battery is widespread and has become a part of daily life with electronic gadgets including cell phones, cameras and laptops, because of its high energy density, fast charging and long cycle life [5,6]. It contains three main components, an anode, cathode and a separator as shown in Figure 1 [1]. A separator is the key component in the battery to prevent it short circuiting and it provides a smooth path for the flow of lithium ions [7,8]. With increasing advancement in technology, the demand for the safer and more environmentally friendly Lithium ion battery is increased. For commercial purposes, the physical and electrochemical aspects of the lion ion batteries should be improved. Because of the separator’s direct connection with electrochemical performance, the separator should have properties such as being thermally stable at high temperatures to prevent the danger of short circuiting, contain interconnected pores less than one micron in size, which could hold the organic electrolyte without leakage, be mechanically robust, hydrophilic in nature and show high affinity towards organic electrolytes, be chemically stable against an electrolyte which is highly corrosive in nature and especially maintain charging and discharging for a long time.

The commercial Polyfin separators for example, PP, PE and trilayer separator (PP/PE/PP), which are manufactured by the dry and wet method, made a promising place in the market. These separators contain high mechanical strength, electrochemical and cycling stability, the capability of continuous operation in ambient conditions without appreciable damage and shut down behavior [2,3] but these Polyfin separators experience some drawbacks such as low ion conduction, ease of meltdown at high temperatures due to low melting temperature (PE-135 °C PP-165 °C) which could result in the thermal runaway of the battery during functioning [4,5], low electrolyte uptake due to hydrophobic material, small lithium transference number, having less stable capacity at high charge density. For replacement of these commercial separators, the single polymer membranes have been prepared. The most commonly used polymers for making the membrane are PVDF-based and its copolymer [6,7,8,9,10,11,12], PVA [13], PVB [14], Polyimide [15], PEO [16], cellulose and its copolymer [17,18], PAN [19,20,21], PDMS [22], PEEK [23] and other polymers because of its high hydrophilicity, high dielectric constant, ease of film forming ability, mechanical strength and high melting temperature. The production method of polymeric membranes is divided into different categories which include the phase inversion method with pores forming such as glycerol [24], PVP [25] and DBP [26], the electrospinning and solvent evaporation method. In spite of these great characteristics of the single polymer membrane described above, these single component membranes still do not fully satisfy the requirements of the lithium-ion battery applications due to a single component as well as manufacturing flaws. For example, the phase inversion method is quite effectively used to make a single component porous structure with submicron size pores, but they do not provide enough support mechanically. In the electrospinning method for making a non-woven separator, the fibers create a network structure which facilitates the membrane with high porosity and high ion conduction but the leakage of electrolytes and capacity fading during charging and discharging make it less compatible for a lithium-ion battery and its mechanical strength is also not sufficient enough. Along with these drawbacks, the single component separator experiences less adhesion to the electrolyte, low ion conduction, less stability against chemical reactions during cycling and loss of structural integrity due to heat generation during the process. These overall disadvantages make the separator less applicable to the lithium-ion battery field. Research scientists have provided many solutions to counter these polymeric membrane problems. One of the solutions to these problems is to incorporate the ceramic nanoparticles, for example, SiO_2_, Al_2_O_3_, MgO, the hydroxides of aluminum and magnesium which are hydrophilic in nature and possess nano and mesopores inside shells for the smooth hauling of lithium ions. The application of the ceramic nanoparticles in the separator is divided into three categories, for example, a ceramic coated separator, a ceramic nanoparticles-filled separator and a ceramic nanoparticles-filled nonwoven membrane [27]. The effects of ceramic nanoparticles on the separator’s physical and electrochemical performance are investigated and a comparison bare separators is also provided.

## 2. Inorganic Particle-Coated Composite Membranes

To protect the surface of the Polyfin separator, the nanoparticles create a network structure on the surface of the substrate with the help of a polymer binder, which enhances porosity. Their hydrophilic nature can provide not only affinity towards electrolytes but also enhance the electrolyte uptake, ion conduction and also play an important role in boosting the cyclic capacity. The refractory property of ceramic nanoparticles and the high melting temperature of the binder can give the separator a thermally stable coating and provide better protection against heating. The most common ceramic nanoparticles used to modify membranes are titania (TiO_2_) [28], silica (SiO_2_) [29,30,31,32], Alumina (Al_2_O_3_) [33,34,35], nickel oxide (NiO) [36]. The other nanoparticles less common in the market are hydroxides of magnesium and aluminum [37]. The most commonly used binders are PVDF [38,39] and its copolymers, Polyimide [40,41,42] and so forth. The ceramic particles are applied by different techniques for example, dip coating including both sides coating and one side coating by automatic machine [37,43,44], coating by sol-gel, coating by plasma treatment and grafting of the nanoparticles on the surface [45], surface modified nanoparticles coating, fibrous substrate use. These categories also discussed in previous reports [46,47].

### 2.1. Dip-Coating Technique

Dip-coating has been employed to create a uniform coating with controllable thickness and large surface area and is widely applicable in commercial and non-commercial sectors [48]. The general recipe to develop the coated separator is as follows: The nanoparticles disperse into an organic solvent by ultra-sonication and then a specific binder is added into the homogeneous solution stirrer for a specific period of time at room temperature or elevated temperature. The separator is dipped into a solution to generate the coated separator. Afterwards, it dries under vacuum at a specific temperature. For example, Shi, C. et al. [49] coated Alumina on a PE separator using a Polyimide binder. The ceramic coated separator improved the thermal resistance and electrochemical characteristics of the PE separator by developing a microstructure with interconnected submicron size pores on its surface. They also investigated whether by using Polyimide binder, more uniform and well dispersed particles coating could be achieved as compared to a PVDF-HFP created coating, due to its not gelling property during the drying process. The hydrophilic nature of Polyimide and alumina nanoparticles enhanced the electrolyte uptake and ion conductivity, which in turn increased the capacity retention. Another example is Dong, X. et al. [50], who coated the zeolite nanoparticles with a different ratio of Si/Al on the surface of the Polypropylene separator by using a PVDF binder. They found that the coating crystallinity decreased with increasing Si/Al ratio in zeolite and an optimum ratio of Si/Al for good performance was 25. The coated separator with 25 ratio Si/Al showed high electrolyte uptake and ion conductivity due to the hydrophilic nature of zeolite and also minimized the interfacial resistance which resulted in a better discharge capacity at a different charge density.

Most of the oxides, including alumina and silica, are produced from their hydroxides in the presence of sodium hydroxide, which could produce many impurities in the particles and could damage the performance of the lithium-ion battery. An approach to resolve this is to adopt the hydrothermal method followed by jet milling to create fine particles without impurities. For example, Lee, D.-W. et al. [51] synthesized ultrafine α-Al_2_O_3_ with the help of jet milling accompanied with the hydrothermal method and coated on the surface of the Polyfin separator with the help of a PVDF-HFP binder. The slurry was mixed with acetone in the ball milling process for 2 h and the PE separator was dipped into the solution and dried in a vacuum oven. They found that the jet-milled processed particles coating with different pulverization pressures from 4 to 6 bar produced small and uniform size particles with 99.99% purity, as compared to un-milled particles and provided high porosity (61–63%), high electrolyte uptake (243–244%), as compared to an un-milled hydrothermal made particle coating, which had 59% porosity and 212% electrolyte uptake. Also, a composite separator with a jet milled refined alpha alumina particle developed high-quality coating. Both types of coating made the Polyfin separator highly thermally stable and provided high electrolyte uptake. The two types of particles delivered the same capacity of 87% after 100 cycles at 0.5 C, which was slightly superior than the PE separator.

### 2.2. Particles Doping Technique

To enhance the compatibility between a binder and ceramic nanoparticles, another approach of researchers is to dope the ceramic nanoparticles with the binder, which boosts the performance of a battery and could provide an option of using an excess amount of particles in the coating with a minimum risk of pore blockage. For example, Luo, X. et al. [52] coated the separator with cerium oxide nanoparticles on the PE separator by using synthesized P(MMA-BA-AN-St) as a doping agent. They investigated the copolymer coating with and without the doping of cerium oxide nanoparticles from 0 to 200 wt. % and found that at 100 wt. % CeO_2_, the porosity value increased to 66% due to large interconnected voids, which created an extra space. They also found that coated copolymer with 10 wt. % CeO_2_ was responsible for better electrolyte uptake which was 81.0 g m^−2^ and better stability was attained at 200 wt. % CeO_2_ content. The copolymer doped with 100 wt. % ceramic nanoparticles enhanced the tensile strength up to 52MP value. The ion conduction value reached 2.5 mS cm^−1^ at 10 wt. % CeO_2_ contents. With the composite separator, 75% and 72% cycling capacity retention was gained with 10 wt. % CeO_2_ and 100 wt. % CeO_2_, respectively, after 200 cycles at 0.1 C rate.

### 2.3. Inorganic-Organic Hybrid Materials Use in Coating

Along with the use of inorganic nanoparticles in coatings, inorganic and organic containing group materials have been developed to make the coating more efficient. Yang, S. et al. [53] coated the nano Polypyrrole/organic montmorillonite- (nano-ppy/OMMT-) on the surface of the PE separator. They found that the electrolyte uptake increased due to PPy/OMMT with the high surface area and high aspect ratio, which also promoted the lithium migration through the composite separator and resulted in high ion conductivity (4.31 mS cm^−1^). They found that the composite coated separator enhanced the thermal stability and the mechanical strength was increased due to adhesion between the organic coating and PE internal surface. The composite coating also reduced the HF contents which were formed due to LIPF_6^−^_ reaction with water and restrained the dissolution of Mn^3+^ from the active cathode material LiNi_1/3_Co_1/3_Mn_1/3_O_2_ at 80 °C. Ultimately, the discharge capacity was superior at high temperatures as compared to the PE separator.

### 2.4. Water Soluble Binder Use in Coating

To avoid the use of an expensive organic solvent for binder and particles in the coating, the severe effects of a solvent on the environment due to its toxicity and partial extraction from the coated surface due to high boiling temperature, the researchers’ approach is to use water-soluble binders to reduce the cost of the coating. PVA [54], cellulose and its derivatives [55,56,57,58,59,60], SBR [61] which has an easy solubility at room temperature and requires less time for preparing the coating, are a few examples. Deng, Y. et al. [62] created a thin coating on the polyethylene (PE) separator by using aluminum oxide ceramic nanoparticles with PVDF-HFP and a carboxyl methyl cellulose (CMC) binder. The CMC polymer provided a better uniform solution and PVDF-HFP acted as colloidal nanoparticles along with alumina and created an extra space on a Polyfin surface, which enhanced the porosity as well as electrolyte storage space. The coated separator also enhanced ion conductivity and thermal stability. The composite separator delivered 87.6% capacity retention at a 0.7 C/1 C charge/discharge condition for more than 300 cycles by using the pouch cell due to the hydrophilic nature of both binders and inorganic ceramic nanoparticles. Jeon, H. et al. [63] used a surfactant and water-soluble CMC binder to coat the alumina particles on the surface of the polyethylene separator. The surfactant’s non-polar group reduced the surface tension between water molecules and the hydrophobic surface of PE and made uniform coating possible. The CMC and alumina coverage provided the surface better ion conduction due to its hydrophilic nature and reduced the interfacial resistance by boosting the lithium migration through the separator. The binder and ceramic nanoparticles’ high melting temperatures also enhanced the thermal stability for 140 °C, and 93.6% capacity retention was gained after 400 cycles at a 0.2 C rate. Yang, C. et al. [64] coated the Boehmite nanoparticles on the surface of the PE separator by using polyvinyl alcohol (PVA) as a binder. The slurry was coated on both sides of the separator by a dip coating process. They found that the thermal stability was improved due to the good affinity between AlOOH and PE and the formation of an interlocking interface structure prevented shrinkage of the separator even at 180 °C. The coated separator minimized the interfacial resistance and the ion conduction value reached 6.56 mS cm^−1^ with a slight improvement in electrolyte uptake and capacity by using LTO/Li coin cell assembly. Another example is Kim, S.W. et al. [65], who coated the alumina nanoparticles on one side of the PE separator by using a mixture of butyl acrylate (BA) and methyl methacrylate (MMA) as a binder along with a liquid oil additive to prevent water damage during storage. They found that the alumina was uniformly dispersed into an aqueous binder and the porous coating was developed which reduced the thermal shrinkage of the PE separator and the hydrophilic nature of the ceramic particles enhanced the electrolyte uptake and ion conduction with low impedance. Also Lee, H. et al. [66] modified the hydrophobic surface of the PE separator and coated the alumina nanoparticles on the surface of the separator with the help of a water-soluble CMC binder. The coating procedure is as follows: the polyethylene surface was pre-treated with a Polydopamine solution. The ceramic aluminum oxide particles were dispersed into a CMC and water solution. Then, the obtained slurry was cast on one side of the polyethylene separator by using a doctor blade and was dried in the fume hood at 70 °C and further dried in the vacuum. They found that the Polydopamine enhanced the hydrophilicity without damaging the morphology of the PE separator, which was evidenced by the enhanced Gurley number. The adhesion strength between the surface and ceramic nanoparticles increased as well as a high ion conduction of 0.758 mS cm^−1^ with a 119.5% electrolyte uptake value achieved with the composite separator. With the coated separator, the charge-discharge capacity increased slightly.

### 2.5. Inorganic Particles Coating on Cathode

The ceramic nanoparticles provide good thermal stability to the Polyfin separators but due to their fragile nature and brittleness make them less mechanically strong and do not stand against winding and assembly of the batteries [26,67,68], so as to avoid the shrinkage and mechanical weakness of the separator, researchers proposed the coating of the particles on the surface of the cathode with the help of a binder which can create a highly porous structure and provided high lithium ion diffusion. It can also remove the further usage of the non-economical Polyfin separator. For example, Chen, P. et al. [69] coated the synthesized zeolite imidazolate frameworks on the surface of the cathode by blade coating with a PVDF binder. As shown in Figure 2a–d, the zeolite created an interconnected and well organized porous structure with a uniform surface which enhanced electrolyte uptake. The good adhesion between the coating and the cathode surface reduced the interfacial resistance by providing a smooth path for lithium ions migration and developed high ion conductivity as compared to the PP separator, which had high resistance due to the hydrophobic surface. The separator was thermally stable as well as exhibiting low interfacial resistance and the high protection of the lithium anode surface due to the trapping of HF content in the electrolyte provided 84.42% capacity retention at a 1 C rate with NCM/Li cell assembly as compared to the PP separator, which had 79.56% capacity in the same condition (Figure 2e,f). They also found that the zeolite coated cathode had the capability to work at higher temperatures with a reasonable capacity without shut down.

Similarly, Mi, W. et al. [70] created a thin coating of alumina nanoparticles on the surface of the lithium titanate (Li_4_Ti_5_O_12_ LTO) electrode by a blade coating with the help of a polyvinyl alcohol (PVA) binder and compared the results with the PP separator and the freestanding PVA/alumina separator. They found that a crack free and micro to macroporous coating with a similar thickness of PP separator was developed on the surface of the electrode without any cracks, which enhanced the porosity and led to high ion conductivity (2.39 mS cm^−1^). The alumina coated separator on the electrode had high thermal stability and was mechanically stronger as compared to the PP separator. With an optimum value of 0.4 wt. % of PVA, the coating delivered a slightly higher capacity as compared to the PP separator with other ratios of binder in the separator due to its low impedance and high affinity for hydrophilic nanoparticles.

### 2.6. Coating by Sol-Gel Technique

In some cases, the inorganic nanoparticles can be attached to the surface using an efficient sol-gel method because solvents which dissolve the binder and disperse the ceramic nanoparticles are usually harmful to the environment and do not provide enough adhesive strength to the coating. So, the researchers’ approach is to mix the precursors of the specific ceramic nanoparticles with a polymer binder and apply on the surface of the Polyfin separator instead of commercial nanoparticles. For example, Feng, G. et al. [71] coated the hydrophobic silica aerogel on both sides of the PE separator. They found that the coated separator contained 3D mesoporous structure with interconnected voids which delivered high ion conduction and high electrolyte uptake was obtained due to the large surface area and the highly porous structure of silica nanoparticles. The 3D network structure of silica coated separator was thermally stable at 160 °C with 75% original structure retention. Chao, C.-Y. et al. [72] coated silica nanoparticles by using the sol-gel technique. The separator was thoroughly rinsed with dopamine solution and dried. The separator was then coated by dipping into TEOS and r-glycidoxy propyl trimethoxy silane, taken out and washed with water and dried. The dopamine created hydroxyl and amine groups on the surface of the substrate due to oxidation and silica nanoparticles were attached to it by a condensation process. They found that the silica coating on the surface created a well packed and uniform interconnected porous structure which enhanced electrolyte uptake, ion conduction and enhanced thermal stability with 22% shrinkage at 160 °C as compared to the PP separator which shrunk by 55% in the same conditions. The coated separator with hydrophilicity showed better electrolyte holding capacity, reduced interfacial resistance and delivered 75% capacity retention at a 2 C rate with excellent columbic efficiency. Zhang, Z. et al. [73] made a slurry of polyvinyl alcohol and silica nanoparticles by using TEOS and coated it on a PP separator. They found that the coating contained thoroughly developed and well-ordered pores when the TEOS amount increased up to 7.5 wt. % in the coating solution. The hydrophilic nature of the silica produced from TEOS enhanced electrolyte uptake (201.2%) and ion conduction (1.26 mS cm^−1^) and good compatibility between particles and the PVA binder provided 100% flexibility to the coated separator without rupture of the coating. The initial discharge capacity was 131.8 mAh g^−1^ and maintained 91.0% capacity retention after 100 cycles at 0.2 C rate as compared to the PP separator (73.6%) due to the retention of electrolytes in the interstitial voids of the coating. Xie, J.-D et al. [74] coated silica nanoparticles on the surface of a trilayer Polyfin separator by sol-gel technique. The amount of silica was controlled by the TEOS precursor. They found that the spherical shape silica nanoparticles were partially covered by submicron pores which led to ion conduction up to 0.44 mS cm^−1^, high electrolyte uptake and stable capacity with the LTO anode. The coated separator enhanced thermal resistance. Chen, W. et al. [75] coated silica nanoparticles on PE separator with help of cellulose diacetate by sol-gel technique. They found that with 9.4 wt. % silica, the coating had a highly porous structure and electrolyte uptake was increased to 277.3% with 0.624 mS cm^−1^ ion conductivity due to the surplus hydroxyl group in the coating. The silica coated layer suppressed the dendrite problem and enhanced the capacity by increasing the safety of the battery.

### 2.7. Polydopamine Use in Coating

In some cases, the Polydopamine can be mixed with other polymers to enhance the attraction of nanoparticles on the surface and obtain an increment in the hydrophilicity of the Polyfin hydrophobic separator. For example, Pi, J.-K. et al. [76] coated the Polydopamine (PDA) and Polyethyleneimine (PEI) mixture on the PP separator before treatment with zirconia nanoparticles. They found that the PDA/PEI coating had a thinner and smooth layer and provided stability as compared to the PDA layer alone. They found that the PEI amino group enhanced the inorganic particles adhesion towards the surface which was responsible for creating a uniform layer on the surface. ZrO_2_ nanoparticles did not increase the thickness without blockage of pores and the hydrophilic nature of the inorganic filler enhanced the electrolyte uptake, ion conduction and had high capacity retention (96.1%) as compared to the PDA coated (94.9%) and bare PP separators (83.3%). Shi, C. et al. [77] coated the alumina nanoparticles on the surface of the PE separator with the help of a CMC/SBR binder by a dip coating process and afterwards hydrophilic Polydopamine was deposited on it by using the chemical vapor deposition method. They found that the ceramic coating created an interconnected porous structure with a submicron pore size on the surface of the PE separator, which enhanced the ion conduction (1.11 mS cm^−1^). Though by using PDA, the composite separator porosity was reduced but it proved helpful in minimizing the thermal shrinkage and thermal stability reached 200 °C with zero dimensional change. The ceramic coating and PDA coating slightly improved the capacity performance.

### 2.8. Inorganic Particles-Coating by Plasma Radiation and Grafting Technique

The particles-coating enhanced the properties of the Polyfin separators but, due to low affinity between particles and substrate, the coating adhesion strength reduced. The coating uniformity as well as enhancing the connection between substrate and ceramic, the research approach is to use plasma radiation method which enhances the surface hydrophilicity of the substrate and nanoparticles. Surface modification by plasma radiation has been successfully employed for commercial applications because it can easily introduce electrolyte affinity and polarized groups on the surface of a hydrophobic Polyfin separator including OH, -COOH, R–O–R′, -NH_2_ groups. It depends on the types of reactive gases such as air, NH_3_, CO_2_ and O_2_ that the inorganic particles will anchor on the surface with the help of the aforesaid groups. One advantage is that Plasma treatment is easily applicable as compared to chemical vapor deposition (CVD) and atomic layer deposition (ALD) which are far from manufacturing processes’ applications. For example, Jeon, H. et al. [78] applied the plasma treatment to attach the alumina particles on the surface of the polyethylene separator and compared the results with a DLSS surfactant assisted ceramic coated separator. As shown in Figure 3a–d, the plasma treatment and surfactant-assisted coating was similar but had a low Gurley number and etching effect of plasma treated coating created wide pores. The hydroxyl groups created by plasma radiation provided a uniform coating and created hydrogen bonding with water-soluble CMC and delivered high uptake and ion conduction. The plasma-treated coated membrane also delivered less impedance resistance as compared to surfactant-based coating due to the higher number of lithium ions transferred through the separator. The coating created by plasma application and surfactant boosted the capacity up to 94.7% for 1000 cycles at 0.2 C (Figure 3e,f), as compared to the bared PE separator which remained stable up to 300 cycles and further dropped to zero with LiMn_2_O_4_ cathode and lithium-based anode.

Along with the plasma technique, the grafting of nanoparticles on the surface of the substrate by electron beam radiation has also been adopted by researchers due to its simplicity and easy modification. The highly intense beam radiation creates free radical which initiate the grafting process with a fast rate [79,80]. For example, Zhu, X. et al. [81] grafted the nano-titania particles onto the surface of the PE separator by the electron beam radiation method. The electron beam radiation created free radicals on the surface of the PE separator which first grafted the agent and then the titania was attached to it with the hydrolysis process. They found that the titania was uniformly distributed on the surface without any blockage of pores and thickness change. The titania nanoparticles provided better thermal stability with minimal shrinkage at 150 °C. The particles coating created a rough surface which enhanced the electrolyte uptake and led towards high ion conduction without increasing the thickness of the PE separator. The titania on the surface provided protection from reacting electrolyte with the anode surface which minimized the risk of damaging the electrochemical window. The grafted titania coated separator had a slightly higher specific capacity. Jiang, X. et al. [82] grafted the alumina nanoparticles on the surface of the PE separator by electron beam radiation. The ultra-thin coating of alumina was created on the surface of the PE separator without any change in thickness. The well-defined porous structure of Al_2_O_3_ enhanced the thermal stability and hydrophilic nature increased the electrolyte uptake, ion conduction (0.53 mS cm^−1^) as well as 85.6% capacity retention after 100 cycles at 1 C rate which was higher than that of the bare PE separator.

Choi, Y. et al. [83] grafted the silica nanoparticles on the surface of a polyethylene separator. The silica nanoparticles made a strong covalent bond with the hydrophobic Polyfin carbon group separator by using electron beam radiation which enhanced the electrolyte uptake and ion conductivity was increased up to 0.82 mS cm^−1^. Due to the covalent bonded coated separator, the thermal stability boosted. The hydrophilic natured silica coated separator showed less oxidation current density in electrolyte due to electron beam radiation and delivered 82.86% capacity retention after 50 cycles as compared to the bare PE separator which maintained at 75.42%.

Similarly, Liu, M. et al. [84] grafted the polyacrylamide on the surface of the Polypropylene and then did a coating of silica particles by sol-gel method. They found that the non-covalent bonding between silica particles and grafted polymer created a uniform coating on the surface with enhanced electrolyte uptake and high ion conductivity (1.43 mS cm^−1^) as compared to the bare PP separator (1 mS cm^−1^). The coated layer of silica and polymer significantly enhanced the thermal stability at 150 °C with just only 12% shrinkage as compared to the bare PP which had a 70% shrinkage.

### 2.9. Surface Modified Particles-Coating

The ceramic nanoparticles have made improvement in lithium-ion battery performance, but their surface still has less attraction towards electrolyte due to less hydroxyl group or any hydrophilic media which need to be improved. So, the surface modification could be one of the options to increase the hydrophilicity of the ceramic nanoparticles in the coating layer. The oxides of ceramic nanoparticles can be hydrolyzed to produce a hydroxyl group on the surface of particles which can create better affinity towards electrolyte and prevent anion interference to make the lithium movement smooth. Also, ceramic particles surface modification is necessary to create a better compatibility between binder and particles adhesion. Here Nho, Y.-C. et al. [85] modified the γ-Al_2_O_3_ ceramic nanoparticles with coupling agent 3-(trimethoxysilyl)propyl methacrylate (TMSPMA). The modified particles were mixed in acetone solution with PVDF-HFP binder and crosslinker (1,3,5-trially-1,3,5-triazine-2,4,6(1H,3H,5H)-trione (TTT) at room temperature and were coated on both sides of the PE separator under humid conditions. The coating was finally exposed to electronic bean radiation for cross-linking. The coupling agent created repulsion in-between particles to avoid agglomeration and a crosslinking agent provided a better settlement of ceramic nanoparticles in the polymer solution. The γ-Al_2_O_3_/PVDF-HFP porous coating on the surface of the PE separator built a more amorphous region which delivered high electrolyte uptake, ion conduction up to 1.3 mS cm^−1^ and the electrochemical window widened up to 5.0 V. After 100 cycles, the discharge capacity was maintained as 99% at a 0.5 C rate due to high ion conductivity.

Cho, J. et al. [86] functionalized the silica nanoparticles with an amino group. The amino group reduced the repulsion gap between electrolyte and silica nanoparticles which enhanced the affinity, and ultimately the hydrophilicity of the nanoparticles was enhanced. The procedure for making a coating solution is as follows: the amino-functionalized silica nanoparticles were dissolved into a DMF solution with PVDF-HFP binder and ball milling was applied for 24 h to make uniform slurry. Then the PE separator was dipped and coated by slurry on both sides. Afterwards, vacuum drying was applied. The coating uniformity and submicron size unified porous structure of the hydrophilic amino-functionalized silica, along with PVDF-HFP on the pristine separator, enhanced the electrolyte uptake, ion conduction up to 0.81 mS cm^−1^ and made separator thermally stable as compared to the PE separator. The amino group on the surface of silica reduced the side reaction between electrolyte and electrode by absorbing the moisture from the electrolyte and less HF contents were produced. The amino-functionalized silica coated separator had 176.3 mAh g^−1^ initial discharge capacity and after 200 cycles the capacity was dropped to 156.0 mAh g^−1^ at a 0.5C rate with 88.5% capacity retention. Another example is Mao, X. et al. [87], who coated synthesized aluminum doped zeolite nanoparticles on the surface of the PE separator and compared the results with the bare PE separator, the zeolite coated and PVDF/SiO_2_ separators. The brief procedure is as follows: A measured amount of synthesized aluminum doped zeolite particles, silica and pure zeolite were mixed in water with PVDF binder, PVP and P123 and coated on both sides of the PE separator. The results in Figure 4a–d show that the aluminum doped zeolite coating provided better results as compared to other coating materials. Well compacted and fully organized nanoparticles on the surface created a porous structure which provided a facile channel to enhance the ions’ movement and ultimately high electrolyte uptake (430.9%) with a 310.3 cc^−1^ gurley value was obtained. The aluminum doped zeolite coating delivered high thermal stability to the Polyfin separator and their interaction between the Lewis acid sites and cation of electrolyte enhanced the lithium transference number by introducing free lithium ions. The coated separator’s ion conduction was improved to 0.54 mS cm^−1^ and it maintained 94.6% capacity retention after 100 cycles at 0.2 C due to low impedance resistance and fast lithium transfer (Figure 4e,f).

### 2.10. Hollow Inorganic Particles-Coating

To enhance the electrolyte uptake by solid ceramic particles, the approach is to convert solid particles into hollow ceramic particles with a single core or multiple cores. For example Liao, H. et al. [88] converted the silica nanoparticles into core-shell particles by emulsion polymerization and coated them on the surface of the Polyfin separator. The core-shell based coating performed dual function to protect the battery by automatically shut down properly beyond 80 °C and also enhanced the thermal stability at the same time which was superior to the PP separator. The coated separator had same ion conductivity and specific capacity as a commercial separator and had no negative effects on electrochemical performance. They found that the coated separator worked at 80 °C without capacity damage.

### 2.11. Different Substrate Use in Coating

To remove the hydrophobic and expensive Polyfin separator substrate for coating, a useful approach proposed by researchers is to coat the ceramic nanoparticles on the other electrospun polymers, which could provide high hydrophilicity and enhance the attributes of the lithium-ion battery and also minimize the cost of the coated membrane. Just like in this case, Zheng, W. et al. [89] coated the silica nanoparticles on the surface of the fibrous PVDF/PAN separator. The ceramic coated separator maintained the structure at 200 °C without any melting. The procedure of making a hybrid separator is as follows: the fibrous membrane was prepared by blending PVDF and PAN in DMF solution and feeding it into the spinning machine. Then the separator was dip coated using pre-mixed PVDF and silica nanoparticles dispersion in acetone (9:1 ratio) and then vacuum dried. The coating of silica nanoparticles created well interconnected and excessive pores on the surface as well as inside the fibrous membrane due to vacuum filtration which provided smooth lithium ions passage and ease of a higher amount of silica application in the coating. The coating with 64 wt. % silica contained high mechanical strength with flexibility, 1.68 mS cm^−1^ ion conductivity value and high electrolyte uptake due to the wide surface area and the hydrophilic nature of silica nanoparticles. The hybrid membrane capacity retention was remarkable with little decay for 100 cycles at a 1 C rate and even at 60 and 120 °C the separator maintained its capacity without damage. Another example of substrate use is Wang, Z. et al. [90] coated the alumina nanoparticles by a spray method on the surface of the tissue paper to decrease the cost of production. They found that the particles had strongly attached to the surface of the paper due to the interaction between particles and cellulose materials’ hydroxyl group. The aluminum oxide covered the large pores of the tissue paper with mesopores or micropores which enhanced the porosity (56%) and uniform structure reduced the lithium dendrite problem and prevented penetration of electrode particles into separator surface. The hydrophilic nature of alumina enhanced the electrolyte uptake which minimized the interfacial resistance that led towards high ion conductivity (1.64 mS cm^−1^) and the initial discharge capacity was 152 mAh g^−1^ with a high initial columbic efficiency of 86.9% and no capacity drop was observed after 60 cycles at 0.5 C. The membrane exhibited 100% restore capacity after different charge densities.

In another example, Wu, D. et al. [91] coated the alumina nanoparticles on the surface of electrospun PVDF fibrous membrane by using PEO as a binder and compared the results with bare PVDF and PE separators. They found that the alumina nanoparticles covered the fibrous separator with help of a PEO solution. The porosity of the coated separator was enhanced but the electrolyte uptake was reduced due to the reduction of pore size as compared to PVDF uncoated separator but was superior than PE separator. The affinity of PVDF and alumina towards electrolyte made high electrolyte uptake and ion conductivity reached up to 2.23 mS cm^−1^. The alumina particles created a self-discharge mechanism in the separator due to the suppression of the dendrite problem and delivered 98.2% discharge capacity retention after 100 cycles at 1C rate as compared to PVDF and PE separators.

A water soluble binder is also applied which makes the coating economical and easy to apply. For example, Xiao, W. et al. [92] Coated zeolite nanoparticle on the surface of the nonwoven PVA membrane with the help of bacterial cellulose pulp as a binder. The ceramic nanoparticles firmly embedded into cellulose fiber which created an interconnected microporous structure on the fibrous PVA membrane. They found that because of the cellulose hydroxyl group as well as the hydrophilic nature of PVA, ceramic particles enhanced the electrolyte uptake (290%). The high uptake and high porosity delivered high ion conduction (2.14 mS cm^−1^) which was superior as compared to the Polyfin separator. The supportive PVA membrane enhanced the strength and reduced the resistance of lithium ion diffusion in a separator. The separator cycling performance was slightly higher than the PE separator after 100 cycles.

Li, W. et al. [93] prepared an alumina coating with the help of a water-soluble sodium polyacrylate (PAAS) binder on the surface of the poly(ethylene terephthalate) PET nonwoven membrane by the dip coating method. They found the three-dimensional porous structure and good affinity of liquid electrolyte towards alumina enhanced the electrolyte uptake, ion conductivity and thermal stability was enhanced with the PET polymer support. Composite separator showed a slight improvement in a capacity value as compared to PE separator. Jiang, F. et al. [94] encapsulated the bacterial cellulose nonwoven separator with silica nanoparticles by the sol-gel method. The hydroxyl group of cellulose was hydrogen bonded with silica nanoparticles which gave it a core-shell structure, also provided rough surface and surplus polar groups which enhanced the electrolyte uptake (145.5%), led towards high ion conduction (18.5 mS cm^−1^) and better cycling performance as well as it was thermally stable at high temperature with zero shrinkage as compared to the PP separator due to unique cross-linked fibrous structure and better electrolyte uptake.

Although most of the research focuses mainly on the solid ceramic nanoparticles but their hydrophilic surface could not accommodate more electrolytes, so the researchers’ approach is to use hollow particles to hold electrolyte more efficiently and enhance the lithium ions movement through the tiny pores of the core-shell particles. Xiao, W. et al. [95] coated hollow silica nanoparticles on the surface of a PET nonwoven mat with the dip coating method. They found that the three-dimensional micron size porous structure was formed on both sides of the fibrous separator and polar nature of hollow silica, PET and PVDF-HFP binder enhanced the electrolyte uptake due to affinity towards electrolyte. The 1:1 ratio of silica and binder coated separator had delivered almost similar discharge capacity as the PE separator at 0.5 C and 1 C rate.

In contrast to the coating method, the sol-gel method is also applied by the researchers to coat the polymer substrate with high uniformity. Just like Luo, D. et al. [96] coated silica nanoparticles using sol-gel method on the surface of the polyphenylene sulfide (PPS) by the dip coating method. They found that the coating converted the disordered porous structure of PPS separator into uniform and proper pore structure. As compared to the PP/PE/PP separator (35%), the composite separator delivered high porosity (56%) and due to a strong polarity of composite separator compounds as well as affinity towards electrolyte, it delivered high electrolyte uptake which led towards high ion conductivity. They found that the high melting temperature of a composite material made it thermally stable structure and due to high electrolyte uptake and ion conductivity, the discharge capacity at high density was superior to the commercial trilayer PP/PE/PP separator. Another example is Xu, Q. et al. [97] coated the alumina nanoparticles on the surface of nanofibrous bacterial cellulose (BC) membrane by the sol-gel method. The hydrophilic alumina particles created a continuous network on the surface of the separator with uniform dispersion, which accommodated high porosity (74.7%) and high electrolyte uptake (625%) as compared to commercial PP/PE/PP separator (82%). They found that the alumina/BC composite membrane’s high uptake and good affinity towards liquid electrolyte led towards high ion conductivity of 4.91 mS cm^−1^ with a high thermal stage at 200 °C without shrinkage. High electrolyte uptake of composite membrane resulted in a large number of lithium ions migration through it which boosted the capacity. At 0.2 C, the initial discharge capacity was 160 mAh g^−1^ which maintained 90% capacity retention after 50 cycles as compared to the bare bacterial cellulose and trilayer PP-PE-PP separators.

### 2.12. Coating by Inorganic Particles Cross Linking Technique

In some cases, the inorganic nanoparticles are cross-linked with the membrane by preventing the blockage of pores using inorganic nanoparticles and enhancing the uniform particles distribution on the surface of a substrate. Park, S.-R. et al. [98] cross-linked the synthesized silica nanoparticles (Figure 5a) on the surface of the fibrous PAN separator by the dip coating method. The short procedure is: the electrospun PAN membrane (Figure 5b) was dipped into a solution of Vinyl-functionalized SiO_2_ with tri (ethylene glycol) diacrylate (TEGDA), which acted as a crosslinker and then the radical polymerization was applied at 80 °C for successful cross-linking (Figure 5c,d). The membrane was free of beads with uniform silica particles’ attachment. They found that the cross-linked membrane had a good affinity towards electrolyte and the polar nature of PAN fiber led to high uptake and ion conduction value was increased up to 2.1 mS cm^−1^. The membrane was thermally stable at 200 °C without any shrinkage and an initial and final discharge capacity was 172.5 and 162.1 mAh g^−1^, respectively, with 94.0% capacity retention by using LiNi_0.6_Co_0.2_Mn_0.2_O_2_ cathode and graphite anode (Figure 5e,f).

All the above methods and techniques provide a high quality coating on the surface of the Polyfin separator which enhances the physical as well as chemical characteristics of it. The ceramic nanoparticles hydrophilic nature, good affinity with electrolyte and heat resistant nature enhance electrolyte uptake, ion conduction and thermal stability of the pristine Polyfin separator. The particle-coated separator could provide better replacement of the commercial separator but somehow, potential risks such as particles detachment from the surface during charging and discharging and pore blockage still exist which could block the ion movement through the membrane which eventually lead toward short circuit.

## 3. Inorganic Particle-Filled Composite Membranes

The inorganic particles filling with polymer membrane make the membrane not only thermally stable but also create extra spaces inside it to smoothen the passage for lithium ions, enhance hydrophilicity and provide sufficient mechanical strength. The inorganic nanoparticles are divided into two categories. One is active fillers which contain lithium salts such as LIBOB, LiDFOB, LiFAP [99,100,101,102], other is passive fillers which consists of inert nanofillers such as TiO_2_, MgO_2_, SiO_2_ and Al_2_O_3_, BaTiO_2_, clays and so forth. [103,104]. The composite membrane was prepared by a simple phase inversion method. The ceramic nanoparticles were dispersed into an organic solvent followed by a specific polymer. Then the solution was coated on a glass plate by the blade and immersed into a coagulation bath or simply dried for extraction of solvent. There is an example of active filler use: Shi, X. et al. [105] prepared a PVDF composite electrolyte membrane incorporated with a different amount of lithium aluminum titanium phosphate glass ceramic (LATP). They found that the lithium salt created a porous structure with the optimum amount (LATP: PVDF 2:1) into a dense structure of PVDF membrane and reduction the crystallinity was attained by increasing fillers amount in it, which resulted in the large amorphous region. They found that at the optimum content value of filler, the electrolyte uptake was 171% and porous structure as well as lithium active filler’s intrinsic conductivity boosted the ion conductivity up to 0.96 mS cm^−1^ at room temperature as compared to bare PVDF separator. The initial discharge capacity of this filler content was 163.5 mAh g^−1^ and maintained a 94.4% capacity retention at a 0.1 C rate and also a stable capacity at a higher charge density rate due to the high migration of lithium ions and low impedance resistance. Similarly, for passive fillers, Costa, C.M. et al. [106] prepared a Polyvinylidene fluoride (PVDF) polymer composite membrane with the incorporation of synthesized silica by way of a non-solvent induced phase inversion method (NIPS) and studied the effect of exposure time in air 1, 5 or 20 min before immersing the membrane into a water coagulation bath. They found that the best results were attained at 1 min exposure time with high qualities of a separator. The composite separator contained an interconnected micropores spherulitic structural morphology and it changed with increasing exposure time in the air before dipping the membrane into the water for phase inversion. With silica nanoparticles, the uptake was increased to 214% and ion conduction value obtained was 0.9 mS cm^−1^. With one minute exposure time, the membrane capacity retention was 79% with 4% loss from 95 mAh g^−1^ that is, an initial capacity value at 2 C.

Wang, X. et al. [107] made a polymer composite membrane using polysulphonamide (PSA) and fumed silica nanoparticles with a simple phase inversion method. The composite membrane contained sponge-like porous structures with a large pore size from 1 to 15 μm after binodal liquid-liquid phase separation [108,109] in the coagulation bath. The membrane delivered high uptake 430% with 0.748 mS cm^−1^. They found that the composite separator was thermally stable at 150 °C without shrinkage due to the high melting temperature of PSA and silica. The membrane delivered 130.5 mAh g^−1^ and after 200 cycles, the discharge capacity remained at 122.9 mAh g^−1^ with 99.6% capacity retention at a 0.5 C rate due to low interfacial resistance and excellent interface compatibility. They found that the composite separator even worked normally at temperature 90 °C at 1 C rate without loss of capacity.

Ali, S. et al. [110] prepared a composite separator containing colloidal alumina and PVDF-HFP polymer by simple phase inversion method. The alumina nanoparticles were treated with citric acid which enhanced its dispersion into an organic solvent and increased their adhesion towards PVDF-HFP in the separator. It also resulted in high strength with a porous structure which had high ion conduction 1.3 mS cm^−1^ at 80 °C and high electrolyte uptake due to oxides on the surface of membrane matrix. The refractory properties of alumina made the separator thermally stable at 150 °C with minimal shrinkage. They found that the colloidal alumina separator enhanced the capacity retention to 95% at 0.2 C rate and even maintained the stable capacity at 140 °C for 200 cycles without appreciable damage.

In addition of single ceramic particle study in composite separators, researchers also used the casting method to create porous composite separators and checked the effect of adding different nanofillers on the separator properties. The recipe for making composite separator is as follows: Different types of ceramic nanoparticle were dispersed into a solvent and then the polymer was mixed into the solution. The slurry was cast on the glass plate and solvent evaporated in the air to get a composite separator. For instance, Nunes-Pereira, J. et al. [111] prepared a composite separator using PVDF-HFP polymer and the effects of nanoparticles montmorillonite, MMT, zeolites, BaTiO_3_ and carbonaceous (multiwalled-carbon nanotubes, MWCNT) addition to the separator was investigated to get the best performance. They found that the homogenous micropores were created after solvent evaporation and different nanofillers created different effects on pore size, electrolyte enhancement, ion conductivity and charge and discharge capacity. These all fillers containing composite membranes shown very stable capacity at 2 C for 50 cycles.

Cui, J. et al. [112] prepared a composite separator containing hydrogen bonded PVDF and cellulose acetate and aluminum hydroxide as a ceramic filler. They found that the sponge-like porous structure developed after solvent non-solvent exchange into a water bath and ceramic hydroxide nanoparticles mixing into polymer reduced the crystallinity, which enhanced the electrolyte uptake consequently leading to high ion conduction 2.85 mS cm^−1^. The polar group on the nanoparticles prevented the anode dendrite problem and composite separator prevented the electrolyte reaction with lithium anode, which was contributed to high charging and discharging with 96.1% capacity retention at 1 C rate.

In the phase inversion method, the thin separator sometimes sticks with the glass plate which cannot be detached easily, so the research approach leads to the use of pore former to enhance the repulsion between plate and polymer material. For example, Zhang, Y. et al. [113] prepared a composite separator containing alumina nanoparticles as a filler and styrene-butadiene rubber (SBR) as a polymer and PEG as a pore former by using the phase inversion method. They found that the pore former proved helpful to avoid the stickiness of the separator on the glass plate during phase inversion, provided a better uniform dispersion of particles in the polymer solution and created a porous structure after removal. The composite membrane possessed high porosity due to interconnected micropores and hydrophilic nature of ceramic particles and capillary forces of pores enhanced the electrolyte uptake, high ion conductivity as compared to PE separator. The membrane was thermally stable at 130 °C because of the heat resistant materials used for making separator. The composite separator slightly improved performance with 156 mAh g^−1^ initial capacity as compared to PE separator and high retention of capacity at different charge densities due to moisture absorption capability of alumina during working of the battery.

As we discussed above, inorganic particles coating with different techniques and inorganic particles filled composite separator, these two methods provided great physical and electrochemical properties but due to thickness increment in coating method by application of inorganic particles, reduction of lithium passage through the separator causing high interfacial resistance, small energy density and low power energy make them less applicable. Also, the coating could be destroyed, and particles could create further resistance by swelling the polymer into the electrolyte due to less adhesive strength. So, the composite separator incorporated with ceramic inorganic particles showed superior performance as compared to coating techniques.

## 4. Inorganic Particle-Filled Non-Woven Mats

The drawback of the traditional non-woven membrane is to have a big pore size and due to insufficient electrolyte holding capacity, great performance loss is expected to happen during long cycling. To resolve the issue, non-woven mats have been prepared by electrospinning method which gives the membrane good affinity towards electrolyte, high uptake and high porosity for easy movement of lithium ions. Inorganic particles filled non-woven are being adopted nowadays because of highly porous structure and higher electrolyte uptake. Electrospinning is common nowadays to make membrane which provides high electrolyte uptake, high porosity and better performance [114,115,116]. For example, Bhute M.V. et al. [117] made an electrospun composite membrane using PVDF polymer and synthesized titania nanoparticles. They found that by incorporation of titania the fibrous structure converted into a porous interconnected structure without agglomeration of particles, caused reduction of crystallinity of PVDF and successfully introduced more amorphous region in the membrane which enhanced the electrolyte uptake to 370%. The good affinity of PVDF and titania towards organic electrolyte boosted the ion conductivity to 4.15 mS cm^−1^.

To avoid the nanoparticles agglomeration and detachment from the fiber, researchers’ useful approach is to add a binder to increase the stability of the ceramic particles inside the membrane during charging and discharging for better performance and good columbic efficiency. For example, Zhang, J. et al. [118] prepared a nanocomposite membrane by using Zeolite, which was chemically bound with a fibrous membrane through a copolymer. The procedure of making membrane is as follows: synthesized fluorinated copolymer PC4SA-co-PMMA-co-PMPS and PVDF were mixed into pre-dispersed Zeolite solution in DMAc/Acetone. The solution was fed into electrospinning machine and membrane was developed, followed by drying in oven. They found that the synthesized copolymer had self-hydrogen bonding with PVDF fiber and created covalent bonding with zeolite particles, which enhanced the dispersibility in nanofibers separator and had good effects on mechanical properties of PVDF membrane. The three-dimensional fibrous structure provided high porosity, thermal stability, high electrolyte uptake 378% which delivered 1.72 mS cm^−1^ ion conduction value due to the high surface area of zeolite and high affinity of PVDF that was higher as compared to PVDF membrane and PP separator. Due to low interfacial resistance, the discharge capacity retention was 93.2%, 93.8% for 200 cycles at 0.2 C and 0.5 C, respectively, which was better than PP and PVDF electrospun separator.

Smith, S.A. et al. [119] made a non-woven ceramic fibrous mat by electrospinning the silica precursor (TEOS). The electrospinning solution contained TEOS, DMF and PAN polymers and an organopolysilazane (OPSZ) precursor. The authors found that the precursor created a connection with pedant chain of TEOS and a ceramic network was made in the fiber, which provided the fibrous membrane mechanical strength and enhanced electrochemical properties. They found that the PAN/ceramic separator had high porosity of 82 ± 5%, 1109 ± 32% electrolyte uptake and 1.04 ± 0.05 mS cm^−1^ for 40 wt. % TEOS contents. The PAN separator with 20 and 40 wt. % had good capacity retention at 0.5 C and 0.2 C as compared to the bare PAN and Celgard separators.

In addition to the ceramic nanoparticles use, there is another approach in which the researchers used organic and inorganic hybrid materials in the form of clay or nanotubes as a filler to collect the hybrid separator with a dual function as they possessed a high surface area, wide aspect ratio and intercalated or exfoliated properties. For example, Fang, C. et al. [120] prepared PVDF fibrous membrane by using inorganic/organic-containing material montmorillonite (MMT) with the help of an electrospinning technique. The fibrous membrane consisted of multilayer fiber with a different diameter which created a three-dimensional porous structure. The prepared membrane delivered high electrolyte uptake 333% and ion conduction was 4.2 mS cm^−1^ with 5 wt. % montmorillonite contents due to polar nature of the PVDF and high surface area of MMT that enhanced the electrolyte affinity towards polymer membrane. They found that the membrane was thermally stable without shrinkage and mechanical strength increased from 1.42 MP (PVDF fibrous membrane) to 3.32 MP with 7 wt. % montmorillonite. The composite separator with 5 wt. % MMT delivered 100% discharge capacity for 50 cycles at 0.2 C. Li, H. et al. [121] prepared the membrane using cellulose and hydroxyapatite nanowires by an electrospinning method and compared the results with PP separator. The cellulose was crosslinked with nanowires by hydrogen bonding and van der Waals force and wires created a network structure with high porosity around fibrous cellulose and enhanced the mechanical strength. Due to high thermal stability of cellulose and hydroxyapatite nanowires, the membrane was found to be heat resistant without shrinkage at 200 °C.

Due to cellulose hydroxyl group, nanowire affinity towards electrolyte and high electrolyte uptake built a strong pathway for migration of lithium ions, which enhanced the ion conductivity of the composite membrane. The low interfacial resistance created high capacity retention (92.3%) at 1C rate and had a capability of working at higher temperature as compared PP separator, which had low electrochemical properties at the same condition.

Zhao, H. et al. [122] prepared a Poly-m-phenyleneisophthalamide (PMIA) three-dimensional structure based electrospun separator with the incorporation of Octaphenyl-Polyhedral oligomeric Silsesquioxane nanoparticles (OPS). They found that OPS nanoparticles reduced the crystallinity of the membrane and the organic part of OPS nanoparticles had a better attraction towards organic polymer and resulted in thin fiber, which enhanced the porosity, electrolyte uptake due to the high surface area of the fiber and mechanical strength due to strong entangled network structure while inorganic part of it provided good thermal and oxidation resistance. With the optimum amount of OPS nanoparticles, the ion conductivity reached up to 1.93 mS cm^−1^. The first discharge capacity value was 157.9 mAh g^−1^ with 89.04% capacity retention after 100 cycles at 0.5 C rate due to low interfacial resistance and high lithium ions’ movement.

Wang, Y. et al. [123] prepared a lightweight thin Polyimide separator with nano-silica by way of the electrospinning method. The SEM results in Figure 6a–d show that the silica particles were uniformly scattered in the fiber without agglomeration and the fibrous membrane contained porous structure which easily made lithium ion transport smooth. They found that the membrane contained high porosity 90%, highly hydrophilic surface with zero electrolyte uptake angle, high uptake value of 2500% due to polar nature of the Polyimide and hydrophilic silica. The composite separator was observed to be thermally stable at 250 °C without shrinkage. They also found that the silica inclusion in the Polyimide membrane made it mechanically strong by regaining its original shape without wrinkles after folding as compared to the Polyimide separator. As shown in Figure 6e,f, due to low interfacial resistance and silica impurity absorbing capability (H_2_O and HF contents), the composite separator capacity retention was 96.3% after 100 cycles at a 5C rate and a high discharge capacity at a different rate as compared to the PP separator. The membrane also delivered stable capacity at 0.2 C at a 50 °C temperature.

The addition of nanoparticles enhanced the mechanical strength of the electrospun separator. However, the mechanical strength is still not sufficient after modifications as compared to commercial separator [124]. 

To resolve this, the researchers proposed substrate use to enhance the mechanical strength of the electrospun composite separator containing ceramic nanoparticles. For example, Liu, J. et al. [125] prepared composite membrane containing ethyl cellulose modified polyethylene separator sandwiched between two layers of electrospun silica and polyamide on both sides. 

The procedure is as follows (Figure 7a): the ethyl cellulose with PVP solution was coated on PE separator and immersed into the water for extraction of PVP and then dried. On the other hand, polyamide and silica nanoparticles were electrospun and two layers of SiO_2_/PI fibrous membrane was attached on both sides of ethyl cellulose (EC) coated PE separator by the thermal calendaring method. As shown in Figure 7c,d, the composite separator had a 3D structure of fiber on the surface of the interconnected porous structure of an EC coated PE separator, which enhanced the porosity to 78% and the polar group of EC and silica nanoparticles acid-base interaction towards electrolyte contributed to high electrolyte uptake and high ion conduction as compared to pristine PE and trilayer separators. They found that the composite membrane was mechanically robust and contained higher mechanical strength as compared to bare PE and trilayer commercial separator due to good adhesive strength in between silica, PI and cellulose. The membrane also delivered high discharge capacity of 162.4 mAh g^−1^ with 83.5% capacity retention after 100 cycles at 0.2 C rate (Figure 7e).

Shen, X. et al. [126] prepared electrospun PVDF-HFP and deposited the ultrathin layer of the alumina nanoparticles on it by using plasma treatment and the atomic layer deposition method. The plasma treatment produced oxygen-derived free radical sites to attach the alumina by condensation reaction (Figure 8a,b). They found that the alumina nanoparticles uniformly encapsulated the PVDF-HFP fiber as shown in Figure 8c–f with large fiber diameter and their electrolyte affinities enhanced the electrolyte uptake, ion conduction and electrochemical stability. Moreover, alumina refractory material properties made the nonwoven separator thermally stable at high temperature. These all properties were superior to the Polyfin PP separator. The alumina wrapped PVDF-HFP showed slightly better capacity retention after 100 cycles at a 1 C rate and less decline in discharge capacity at different discharge rates (Figure 8g,h) with LiMn_2_O_4_/Li cell assembly.

To avoid the agglomeration of nanoparticles in the nonwoven separator, researchers mixed the nonwoven separator with the precursor to get ceramic modified membrane. For example, Jiang, Y. et al. [127] prepared a composite separator with the mixing of PVP and titania by the sol-gel method and an applied electrospinning technique. The separator exhibited automatic off function at 60 °C and regained capacity after temperature got back to the ambient state. They found that the electrospun membrane contained micropores with the surface area of 6.67 m^2^ g^−1^ and interconnected porous structure inside fiber tube enhanced the electrolyte uptake, which maintained the high ion conductivity as compared to PP separator. The PVP/TiO_2_ fibrous separator was thermally stable at high temperature due to thermally stable titania and delivered high discharge capacity at 0.2 C with minimal change for 100 cycles.

There is another example Yanilmaz, M. et al. [128] prepared SiO_2_/PAN hybrid nanofiber membranes by combining the sol-gel and electrospun techniques. The results showed that the porous fibrous structure was developed without any beads. They found the porosity and uptake increased when the silica amount increased from 0 to 27 wt. % in the composite separator and good electrolyte uptake led to high ion conductivity (2.6 mS cm^−1^) due to low resistance and affinity towards electrolyte. The composite separator with different silica contents was thermally stable due to the high melting temperature of PAN and ceramic particles. They also found that the initial discharge capacity was 163 mAh g^−1^ for 27 wt. % as compared to PP separator (155 mAh g^−1^) and maintained the capacity for 50 cycles with almost zero loss at 0.2 C rate.

## 5. Summary and Future Directions

The main focus of study here is the separator, which separates the anode and cathode electrode inside the lithium-ion battery to protect it from an internal short circuit and provide better safety. Some prominent attribute of the separator should be that it must be physically and mechanically strong without any reaction with electrolyte, be responsible to keep lithium ions movement continuous without shrinkage or structure damage, high porosity and interconnected well-groomed porous structure which provide smooth exchange of ions from anode to cathode, hydrophilic nature of the separator material for containment of the liquid electrolyte thoroughly and quickly, high ion conduction, fast and smooth charging and discharging and maintaining the long cycling capacity. The abovementioned attributes somehow can be achieved by a single component microporous separator such as PE, PP, PP/PE/PP.

They are mechanically fit but still inappropriate electrochemically for future batteries. Other developed single polymeric membranes such as PVDF, PVP, cellulose and its copolymer and Polyimide, and so forth, are made by phase inversion method, casting technique and electrospinning using a different pore-forming agent (e.g., glycerol, DBP and PEG). These polymers having a hydrophilic nature, sheet forming capability and solubility into organic solvent, make them applicable for the lithium-ion battery but the mechanical aspect still needs to be improved. To remove the fatal flaws from the separator, researchers’ approach is to add ceramic inorganic nanoparticles to enhance the electrochemical and physical characteristics of the lithium-ion battery. The common inorganic nanoparticles are alumina, titania and silica, which have the ability to disperse into an organic solvent, have a high surface area, hydrophilic nature and thermal characterization. They provide polymer membrane mechanical strength, improve their electrolyte uptake, provide electrochemical stability and enhance the long cycling capacity. The common categories for modification of membrane are coating them on separator surface, inorganic-filled composite and inorganic-filled nonwoven separator. The coating method further includes dip coating, sol-gel method, plasma treatment and grafting, other substrate use and particle surface modification. The dip coating method is common and more practical because of its easy method but it contains some organic solvent involvement which enhances the cost. Sol-gel is another technique in which we somehow reduce the cost of the coating by eliminating the solvent. To remove the substrate hydrophobicity, Polydopamine can be applied on the surface before or after inorganic particles coating to enhance the hydrophilicity of the separator substrate or coating. Plasma technique is applicable for grafting the nanoparticles onto the surface to enhance the strength of the coating and attain uniformity throughout coating. Sometimes the inorganic particles are modified to enhance their hydrophilicity or cross-linking before application for coating. The inorganic nanoparticles coating is applied on the electrospun separator to reduce the cost of the coating substrate. These methods are widely applied to coat the nanoparticles on the surface of separators, which enhance the thermal stability and performance of the lithium ion batteries but increase the interfacial resistance due to increase in the thickness and there are chances of detachment of the nanoparticles during working of a lithium-ion battery due to binder swelling in the electrolyte. The inorganic nanoparticles blended with polymer composites and non-woven separators can remove the above-mentioned coating problem and enhance the battery performance as they contain high porosity, mechanical strength and electrochemical stability as well as stability for long cycling (Table 1). However, some flaws are still present in these inorganic modified separators like organic electrolyte use, which are volatile in nature and could cause shut down of batteries easily, particles detachments from polymers, which could block the pores and cease the lithium transfer and low mechanical strength due to high porosity which could enhance the chance of structure collapse during assembly. These drawbacks could be minimized by enhancing the polymer and particles affinity towards each other and towards electrolyte for better capacity. To minimize the cost and solve the mechanical stability problem, the solid electrolyte separator could be used because they reduce the components of the lithium-ion batteries by excluding usage of the flammable organic electrolyte and enhance the safety of the batteries without minimizing the performance.

## Figures and Tables

**Figure 1 membranes-09-00078-f001:**
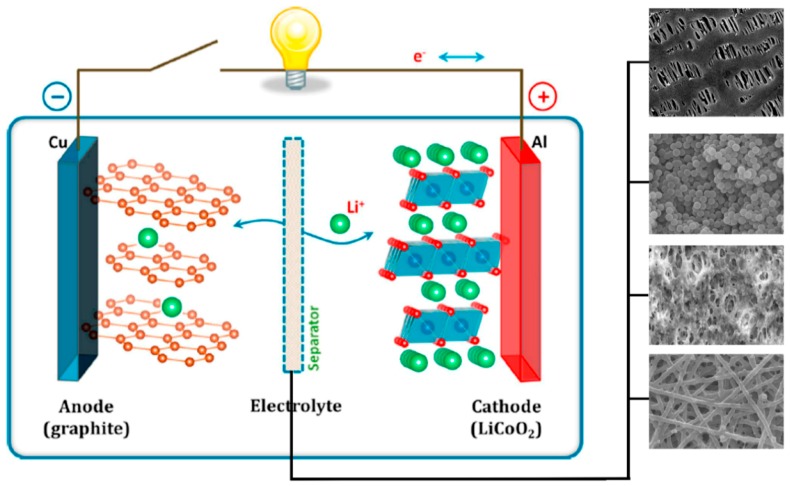
Schematic of Li-ion battery mechanism [1], with copyright permission from American Chemistry Society.

**Figure 2 membranes-09-00078-f002:**
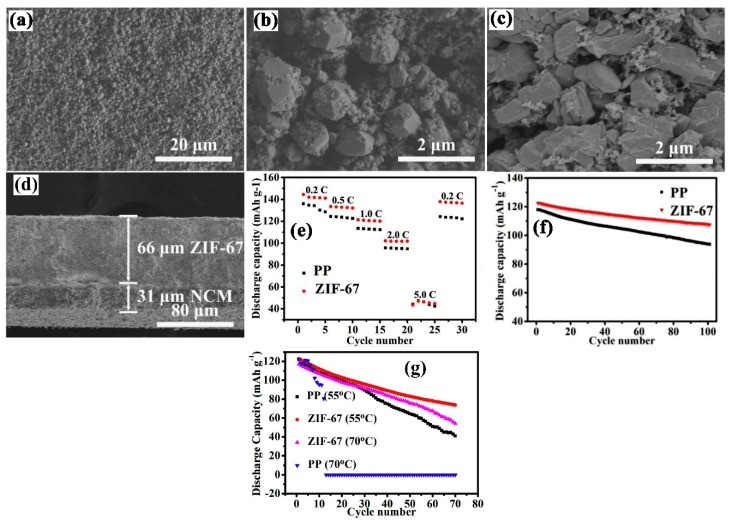
(**a,b**) Top view of ZIF-67 separator with different magnifications. (**c**) top surface image of NCM electrode. (**d**) cross-sectional view of ZIF-67 separator. (**e**) discharge capacities of the PP separator and ZIF-67 separator at different C rates. (**f**) discharge capacities of the PP separator and ZIF-67 separator at the rate of 1.0 C. Inset (**f**) discharge capacities of the PP and ZIF-67 separator under 55 and 70 °C at 1.0 C rate (**g**) Discharge capacity of the PP separator and the ZIF-67 separator at 1.0 C under 55 and 70 °C. [69], with copyright permission from Elsevier.

**Figure 3 membranes-09-00078-f003:**
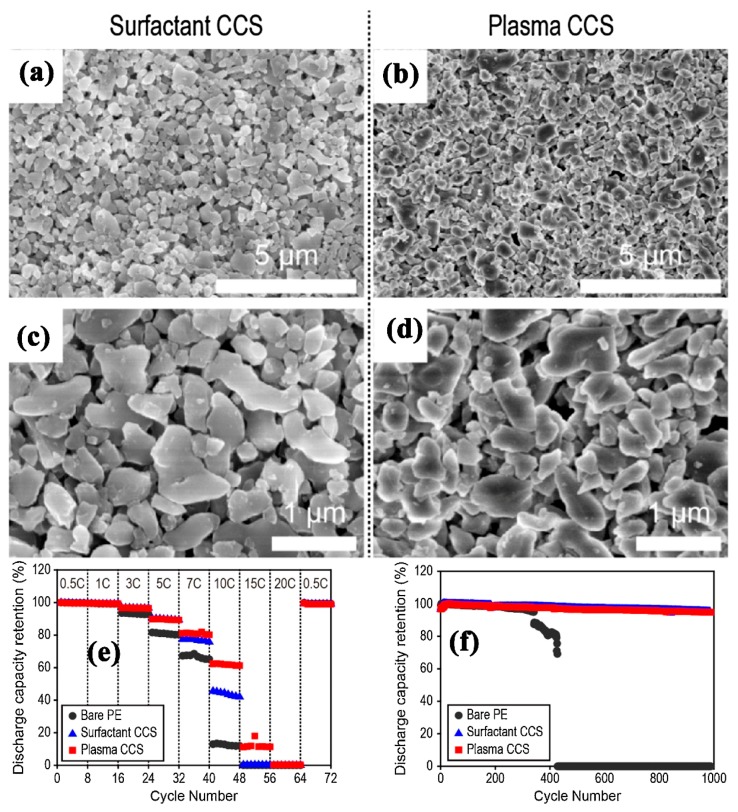
Top view of (**a,b**) surfactant treated (**b,d**) plasma treated Al_2_O_3_ ceramic-coated PE separators (**e**) discharge capacity at different C rate and (**f**) discharge capacity of the unit cells for 1000 cycles [78], with copyright permission from Elsevier.

**Figure 4 membranes-09-00078-f004:**
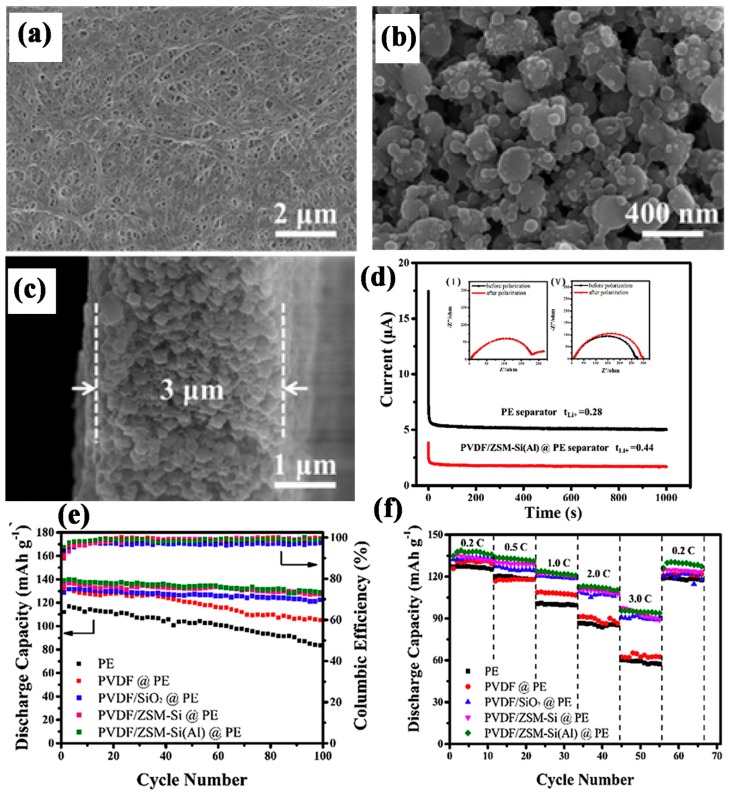
Top view of (**a**) PE (**b**) PVDF/ZSM-Si(Al)@PE (**c**) cross sectional view of PVDF/ZSM-Si(Al)@PE (**d**) chronoamperometry of Li/PVDF/ZSM-Si(Al)@PE separator/Li assembled cells at 25 °C (**e**) discharge capacities and corresponding columbic efficiencies of different composite separators bare PE separators at 0.2 C rate (**f**) different discharge C-rate capabilities of different separators [86], with copyright permission from Elsevier.

**Figure 5 membranes-09-00078-f005:**
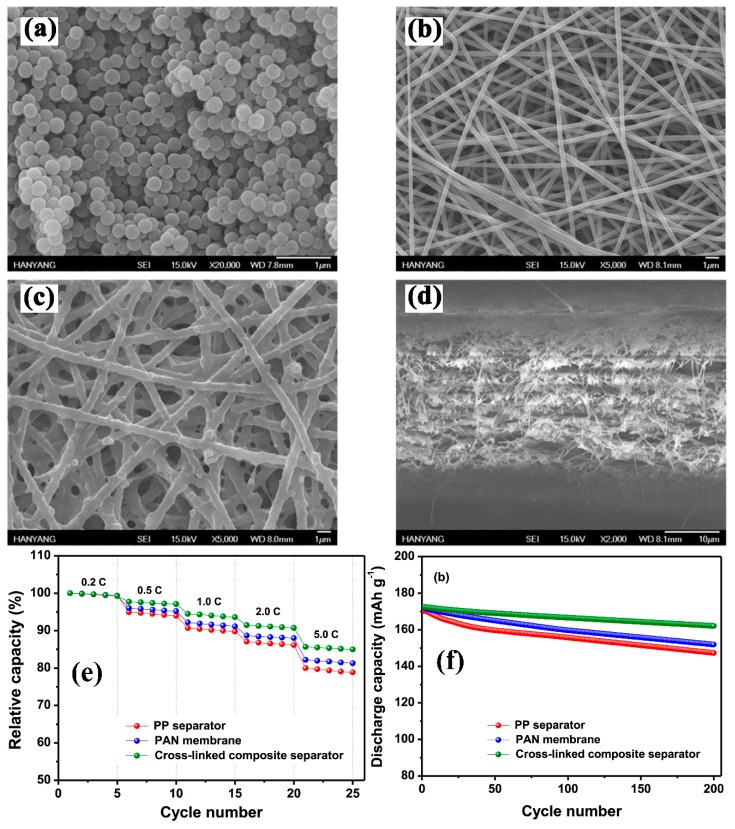
(**a**) SEM of reactive SiO_2_ particles, (**b**) Top view of PAN fiberous separator, and (**c**) composite separator cross linked with SiO_2_ (**d**) cross-sectional view of the crosslinked composite separator (**e**) discharge capacities of different separators at different C rates (ambient temperature) (**f**) cycling capacities of separator a 0.5 C rate for 100 cycles [98], with copyright permission from Elsevier.

**Figure 6 membranes-09-00078-f006:**
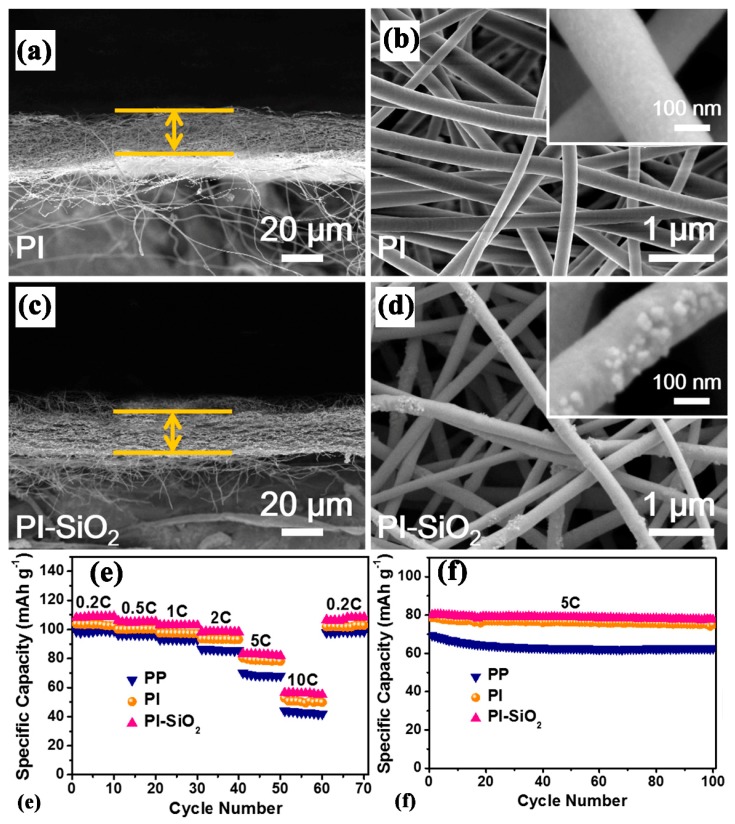
(**a**) Cross-sectional and (**b**) surface SEM image of PI (the inset is corresponding high-magnification image) (**c**) cross-sectional and (**d**) surface SEM image of PI-SiO_2_ (the inset is corresponding high-magnification image) (**e**) rate capability with PP, PI and PI-SiO_2_ separators (0.2–10 C) (**f**) cyclic performance at 5 C at room temperature [122], with copyright permission from Elsevier.

**Figure 7 membranes-09-00078-f007:**
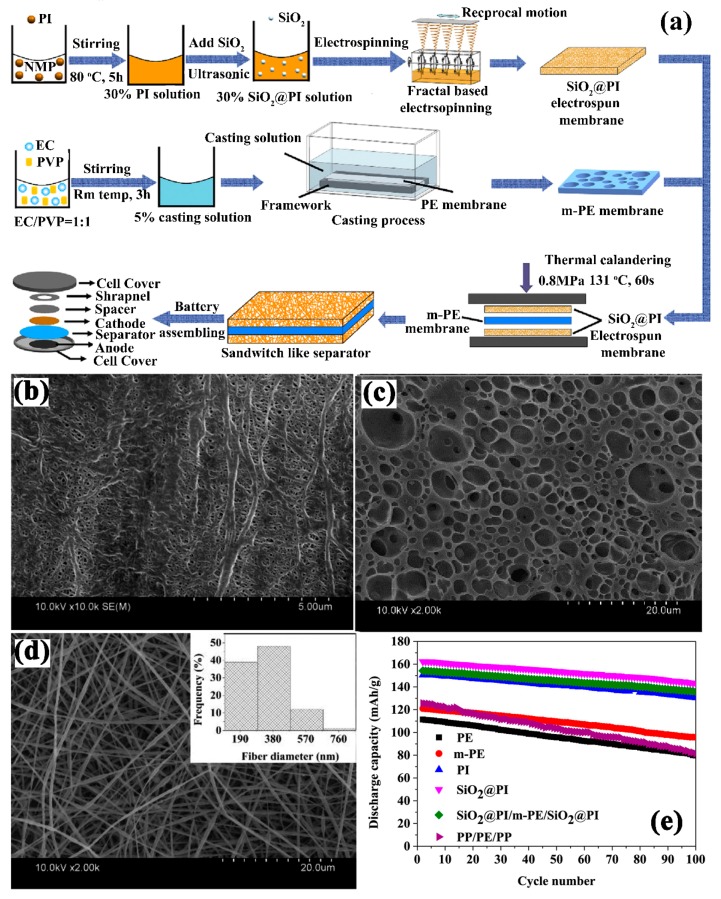
(**a**) The schematic diagram of composite separator preparation and battery assembly process SEM view of (**b**) PE membrane, (**c**) m-PE membrane (**d**) SiO_2_@PI containing 2% SiO_2_ (**e**) discharge capacities of different separators at 0.2 C rate [125], with copyright permission from Elsevier.

**Figure 8 membranes-09-00078-f008:**
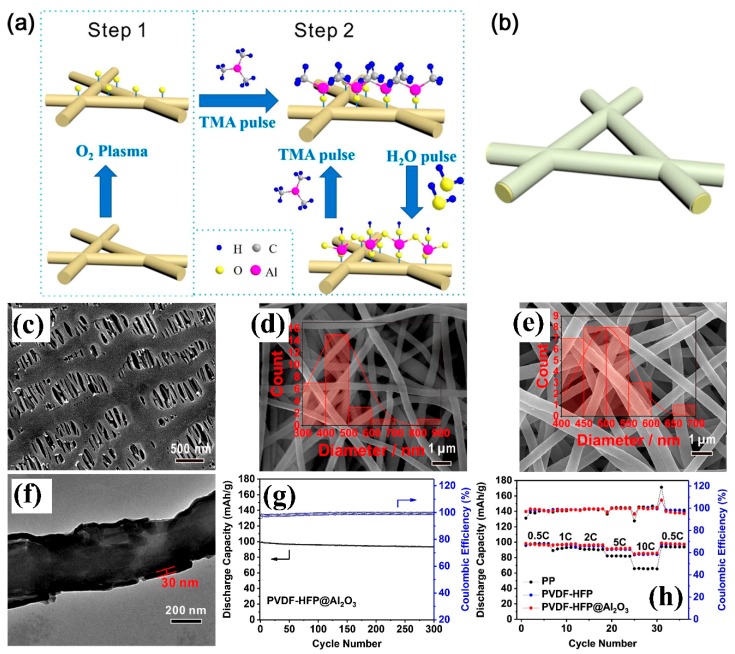
Schematic diagram depicted (**a**) the principle and the fabrication steps and (**b**) the final structure of ALD Al_2_O_3_ core shell nonwoven separator. Top view of (**c**) PP separator, (d) PVDF-HFP fibrous membrane and (**e**) PVDF-HFP@ Al_2_O_3_. (**f**) TEM image of the PVDF-HFP@Al_2_O_3_ fibrous membrane. (**g**) long cycling battery performance assembled with PVDF-HFP@Al_2_O_3_ separator (**h**) discharge capacities of different separators at different C rates [126], with copyright permission from Elsevier.

**Table 1 membranes-09-00078-t001:** Additional information’s about composite membranes.

Substrate/Inorganic Particles	Polymer Binder	Fabrication Method	Composite Separator Thickness (μm)/Electrolyte	Ion Conduction (σ)/Thermal Stability/Shrinkage Percentage	Cathode/Anode	Electrochemical Performance	Refs.
PE/Al_2_O_3_	Polyimide	Automatic coating machine	26/-	0.70 mS cm^−1^/160 °C/0%	LiMn_2_O_4_/Li	**In 3.0 V to 4.2 V** 0.5 C ≈ 107 mAh g^−1^ 1 C ≈ 105 mAh g^−1^ 2 C ≈ 101 mAh g^−1^ 5 C ≈ 92.5 mAh g^−1^	[49]
PP/Zeolite	PVDF	Dip coating	22/1 M LiPF_6_^−^ EC-DEC-DMC (1:1:1 V:V:V)	-	Li_4_Ti_5_O_12_/Li	Enhanced electrolyte uptake, electrolyte retention and stable capacity at different rate.	[50]
PE/α-Al_2_O_3_	-	Dip coating	32–35/1 M LiPF_6_^−^ EC-DEC (1:1 V:V)	-/140 °C/60%	LiCoO_2_/Li	**In 3.0 to 4.3 V** Thermally stable, 244.4% electrolyte uptake, 87% capacity after 100 cycles for 0.5 C rate.	[51]
PE/CeO_2_	P(MMA-BA-AN-St)	One side coating	75/1M LiPF_6_^−^ EMC-EC-DEC (5:3:2 V:V:V)	-/2.5 mS cm^−1^/135 °C/0%	LiNi_0.5_Mn_1.5_O_4_/Li	**In 3.0 to 5.0 V** 0.2 C ≈ 131 mAh g^−1^ 0.2 C ≈ 125 mAh g^−1^ 0.5 C ≈ 120 mAh g^−1^ 1 C ≈ 119 mAh g^−1^	[52]
PE/Al_2_O_3_	BA+ MMA	Automatic bar coating machine	16/1 M LiPF_6_^−^ EC-DEC (1:1 V:V)	-/0.68 mS cm^−1^/130 °C/1.6%	LiCoO_2_/Graphite	**In 3.0 to 4.2 V** 0.2 C ≈ 132 mAh g^−1^ 0.5 C ≈ 125 mAh g^−1^ 1.0 C ≈ 119 mAh g^−1^ 2 C ≈ 101 mAh g^−1^ 5 C ≈ 53 mAh g^−1^	[65]
PE/Nano-ppy/OMMT	PVDF	Dip coating	-/1 M LiPF6 DEC-EC-EMC (1:1:1 V:V:V)	-/4.31 mS cm^−1^/80 °C/0%	LiNi_1/3_Co_1/3_Mn_1/3_O_2_/Li	**In 2.7 to 4.2 V** 0.5 C ≈ 125.9 mAh g^−1^ After 100 cycles At 80 °C 80% capacity retention	[53]
PE/Al_2_O_3_	PVdF-HFP/CMC	Dip coating	12/1 M LiPF_6_^−^ EC-DEC-PC-EMC (2:3:1:3 V:V:V)	9.3 mS cm^−1^/110 °C/0%	LiCoO_2_/graphite	**In 3.0 to 4.35 V** 0.2 C ≈ 162mAh g^−1^ 0.5 C ≈ 161mAh g^−1^ 1.0 C ≈ 158mAh g^−1^ 1.5 C ≈ 153mAh g^−1^ 2 C ≈ 144mAh g^−1^	[62]
PE/Al_2_O_3_	CMC	Bar coating on one side	-/1 M LiPF_6_^−^ EC-EMC (3:7 V:V)	0.846 mS cm^−1^/140 °C/0%	LiMn_2_O_4_/Graphite	**In 3.0 to 4.3 V** 0.5 C ≈ 109 mAh g^−1^ 1.0 C ≈ 108 mAh g^−1^ 2.0 C ≈ 99 mAh g^−1^ 5.0 C ≈ 90 mAh g^−1^ 7.0 C ≈ 80 mAh g^−1^ 10 C ≈ 70 mAh g^−1^ 15 C ≈ 50 mAh g^−1^ 20 C ≈ 30 mAh g^−1^	[63]
PE/AlOOH	PVA	One side dip coating	17.15/1 M LiPF_6_^−^ EC-DEC (1:1 V:V)	6.56 mS cm^−1^/180 °C/3%	Li_4_Ti_5_O_12_/Li	**In 1.0 to 2.5 V** 0.2 C ≈ 152mAh g^−1^ 0.5 C ≈ 150mA hg^−1^ 1.0 C ≈ 142mAh g^−1^ 2.0 C ≈ 130mAh g^−1^	[64]
NCM Cathode/Zeolitic imidazolate	PVDF	Blade coating	66/1 M LiPF_6_^−^ EC-DMC-DEC (1:1:1 V:V:V)	1.64 mS cm^−1^/100 °C/0%	LiNi_0.5_Co_0.2_Mn_0.3_O_2_(NCM)/Li	**In 3.0 to 4.2 V** 0.2 C ≈ 142 mAh g^−1^ 0.5 C ≈ 133 mAh g^−1^ 1.0 C ≈ 121 mAh g^−1^ 2.0 C ≈ 102 mAh g^−1^ 5.0 C ≈ 44 mAh g^−1^	[69]
LTO cathode/Alumina	PVA	Blade coating	25/1 M LiPF_6_^−^ EC-DEC-DMC (1:1:1, V:V:V)	2.39 mS cm^−1^/120 °C/0%	LTO/Li	**In 1.0 to 2.5 V** 0.2 C ≈ 171 mAh g^−1^ 0.5 C ≈ 165 mAh g^−1^ 1.0 C ≈ 150 mAh g^−1^ 2.0 C ≈ 0 mAh g^−1^	[70]
PP/SiO_2_	PVDF	Dip coating	-/1 M LiPF_6_^−^ EC-DMC (1:1 V:V)	0.63 mS cm^−1^/160 °C/24.5%	LiFePO_4_/Li	**In 2.0 to 4.3 V** 0.2 C ≈ 158 mAh g^−1^ 0.5 C ≈ 150 mAh g^−1^ 1.0 C ≈ 141 mAh g^−1^ 2.0 C ≈ 133 mAh g^−1^ 4 C ≈ 123 mAh g^−1^ 8 C ≈ 101 mAh g^−1^	[71]
PP/SiO_2_	r-glycidoxy propyl trimethoxy silane	Dip coating	-/-	0.55 mS cm^−1^/160 °C/25%	LiCoO_2_/Graphite	**In 2.5 to 4.2 V** 0.2 C ≈ 5.1mAh g^−1^ 0.5 C ≈ 4.5mAh g^−1^ 1.0 C ≈ 4mAh g^−1^ 2 C ≈ 2.5mAh g^−1^	[72]
PP/SiO_2_	PVA	Dip Coating	-/1 M LiPF_6_^−^ EC-DMC-DEC (1:1:1, V:V:V)	-/1.26 mS cm^−1^/170 °C/8.3%	LiCoO_2_/Li	**In 3.0 to 4.2 V** 0.2 C ≈ 131.8 mAh g^−1^ 0.5 C ≈ 131 mAh g^−1^ 1.0 C ≈ 129 mAh g^−1^ 2.0 C ≈ 119 mAh g^−1^	[73]
PP/PE/PP/SiO_2_	PVDF-HFP	Dip coating	-/1 M LiPF_6_^−^ EC-PC-DMC (1:1:1 V:V:V)	-/180 °C/38%	LTO/Li	**In 1 to 3 V** 0.1 C ≈ 161 mAh g^−1^ 1 C ≈ 152 mAh g^−1^	[74]
PE/SiO_2_	Cellulose diacetate	Coating by Sol-gel method	16/1 M LiPF_6_^−^ EC-DEC-EMC (1:1:1 W:W:W)	-/0.624 mS cm^−1^/-	LiCoO_2_/Li	**In 3.0 to 4.2 V** 0.2 C ≈ 153 mAh g^−1^ 0.5 C ≈ 141 mAh g^−1^ 1 C ≈ 120 mAh g^−1^ 2 C ≈ 91 mAh g^−1^ 3 C ≈ 79 mAh g^−1^	[75]
PE/Al_2_O_3_	CMC	One side Dip coating	27/1.15 M LiPF_6_^−^ EC-EMC (3:7 V:V)	0.758 mS cm^−1^/140/0%	LiMn_2_O_4_/graphite	**In 3.0 to 4.4 V** 86% capacity retention after 100 cycles at 1 C.	[66]
PE/Al_2_O_3_	CMC	Dip coating	24/1 M LiPF_6_^−^ EC-DMC-DEC (1:1:1 V:V:V)	0.71 mS cm^−1^/200 °C/0%	LiMn_2_O_4_/Li	**In 3 to 4.2 V** 0.5 C ≈ 106 mAh g^−1^ 1 C ≈ 104 mAh g^−1^ 2 C ≈ 97.5 mAh g^−1^ 4 C ≈ 85 mAh g^−1^	[77]
PP/ZrO_2_	-	Dip coating	-/1 M LiPF_6_^−^ EC-DEC-DMC (1:1:1 V:V:V)	1.61 mS cm^−1^/140 °C/0%	LiFePO_4_/Li	**In 3.0 to 4.2 V** 0.1 C ≈ 159.4 mAh g^−1^ 0.2 C ≈ 150 mAh g^−1^ 1.0 C ≈ 137.5 mAh g^−1^ 2.0 C ≈ 125 mAh g^−1^ 5.0 C ≈ 100 mAh g^−1^	[76]
PE/Al_2_O_3_	CMC	a wire bar (Mayer bar)-coating process using	26/1M LiPF_6_^–^ EC-EMC (1:1 V:V)	0.967–1.182 mS cm^−1^/140 °C/0%	LiMn_2_O_4_/Li	**In 3.0 to 4.4 V** No shrinkage at 140 °C Stable capacity for 100 cycles as compared to bare PE separator	[78]
PE/TiO_2_	-	Grafting by electron beam radiation	-/1M LiPF_6_^–^ EC-DEC-DMC (1:1:1 V:V)	0.32–0.50 mS cm^−1^/150 °C/36%	LiFePO_4_/Graphite	**In 3.0 to 4.0 V** No shrinkage at 150 °C Slightly better capacity retention as compare to PE separator	[81]
PE/Al_2_O_3_	-	Grafting by electron beam radiation	16/1M LiP_6_^–^ EC-DEC-EMC (1:1:1, V:V:V)	0.53 mS cm^−1^/150 °C/0%	LiFePO_4_/graphite	**In 2.0 to 4.0 V** 0.2 C ≈ 125 mAh g^−1^ 0.5 C ≈ 121 mAh g^−1^ 1 C ≈ 115 mAh g^−1^ 2 C ≈ 102 mAh g^−1^ 5 C ≈ 80 mAh g^−1^	[82]
PE/SiO_2_	-	Coating by grafting	1M LiPF_6_^–^ EC-DEC- EMC (1:1:1, v:v:v)	0.8164 mS cm^−1^/150 °C/0%	LiCoO_2_/Graphitized mesocarbon microbead	**In 3 to 4.3 V** 0.2 C ≈ 149 mAh g^−1^ 0.5 C ≈ 146 mAh g^−1^ 1 C ≈ 142 mAh g^−1^ 1.5 C ≈ 135 mAh g^−1^ 2 C≈130 mAh g^−1^ 3 C ≈ 89 mAh g^−1^ 4 C ≈ 60 mAh g^−1^ 5 C ≈ 45 mAh g^−1^ 7 C ≈ 29 mAh g^−1^	[83]
PP/SiO_2_	-	Grafting and dip coating	28/-	1.43 mS cm^−1^/150 °C/12%	-	0.2 C ≈ 175 mAh g^−1^ 0.5 C ≈ 164 mAh g^−1^ 1 C ≈ 130 mAh g^−1^ 2 C ≈ 85 mAh g^−1^	[84]
PE/Al_2_O_3_	PVDF-HFP	Dip coating and after electron beam radiation	-/1M LiClO_4_ EC-DEC (1:1 V:V)	1.3 mS cm^−1^/-/0%	LiCoO_4_/MCMB graphite	**In 3.0–4.2 V** 99% capacity retention after 100 cycles at 0.5 C rate.	[85]
PE/SiO_2_	PVDF-HFP	Dip coating	25/1.15M LiPF_6_^–^ EC-EMC (3:7, V:V)	0.81 mS cm^−1^/130 °C/0%	LiNi_1/3_CO_1/3_Mn_1/3_O_2_/(MCMB) graphite	**In 3.0 to 4.5 V** Stable cycling performance at 0.5C rate with minimal drop of capacity after 200 cycles.	[86]
PE/Al-SiO_2_	PVDF	Dip coating	18/1M LiPF_6_^–^ EC-EMC-DEC (1:1:1, V:V:V)	0.30–0.54 mS cm^−1^/130 °C/0%	LiCoO_2_/Li	**In 3.0 to 4.2 V** High lithium transference number 0.30-0.44. 95% capacity retention after 100 cycles at 0.2 C	[87]
PP/PS-b-PBA@SiO_2_	HEC	Spray coating	-	0.65 mS cm^−1^/160 °C/2%	LiFePO_4_/Li	**In 2.75 to 4.2** 0.5 C ≈ 145 mAh g^−1^ 1 C ≈ 140 mAh g^−1^ 5 C ≈ 134 mAh g^−1^ 10 C ≈ 120 mAh g^−1^	[88]
PVDF/PAN/SiO_2_	PVDF	Dip coating	-/1 M LiPF_6_^−^ EC-DMC-EMC (1:1:1, V:V:V)	1.50–1.68 mS cm^−1^/200 °C/0%	LiFePO_4_/Li	**In 2.5 to 4.5 V** 0.2 C ≈ 151 mAh g^−1^ 0.5 C ≈ 147 mAh g^−1^ 1 C ≈ 138 mAh g^−1^ 2 C≈126 mAh g^−1^ 5 C ≈ 104 mAh g^−1^	[89]
Cellulose paper/Al_2_O_3_	CMC/PEG	Spray coating	48/1 M LiPF_6_^−^ EC-DEC (1:1 W:W)	1.64 mS cm^−1^/130 °C/0%	LiCoO_2_/Graphite	**In 3.0–4.35 V** 0.5 C ≈ 152 mAh g^−1^, 1 C ≈ 150 mAh g^−1^ 2 C ≈ 142 mAh g^−1^ 4 C ≈ 123 mAh g^−1^ 8 C ≈ 95 mAh g^−1^	[90]
PVDF/Al_2_O_3_	PEO	Dip coating	74/1 M LiPF_6_^−^ EC-DEC-DMC (1:1:1 W:W:W)	2.23 mS cm^−1^/140 °C/2%	LiMn_2_O_4_/graphite	**In 0.2–4.2 V** 0.5 C ≈ 121 mAh g^−1^ 1.0 C ≈ 119.1 mAh g^−1^ 2.0 C ≈ 115 mAh g^−1^ 4.0 C ≈ 110 mAh g^−1^	[91]
PVA/ZrO_2_	Bacterial cellulose	Deposition method	25/1 M LiPF_6_^−^ EC-DEC (1:1, V:V)	2.14 mS cm^−1^/150 °C/3 %	LiFePO_4_/Li	**In 2.5–4.2 V** 0.5 C ≈ 144 mAh g^−1^ 2.0 C ≈ 117 mAh g^−1^ 8.0 C ≈ 81 mAhg^−1^ 32 C ≈ 48 mAhg^−1^ 64 C ≈ 20 mAhg^−1^	[92]
PET/Al_2_O_3_	PAAS	Dip coating	25/1 M LiPF_6_^−^ EC-EMC-DEC (1:1:1 V:V:V)	1.13 mS cm^−1^/150 °C/0 %	LiNi_1/3_Co_1/3_Mn_1/3_O_2_/LiCoO_2_/Li	**In 3.0 to 4.3 V** 1 C ≈ 140 mAh g^−1^ 2 C ≈ 130 mAh g^−1^ 5 C ≈ 112 mAh g^−1^ 10 C ≈ 97 mAh g^−1^ 15 C ≈ 80 mAh g^−1^ 20 C ≈ 64 mAh g^−1^	[93]
BC/SiO_2_	-	Sol-gel coating	76.1/1 M LiPF_6_^−^ EC-DEC (1:1 W:W)	18.5 mS cm^−1^/200 °C/0 %	LiFePO_4_/Li	**In 2.50–4.20 V** 0.2 C ≈ 142 mAh g^−1^ 0.5 C ≈ 120 mAh g^−1^ 1 C ≈ 105 mAh g^−1^ 2 C ≈ 90 mAh g^−1^ 3 C ≈ 70 mAh g^−1^ 5 C ≈ 50 mAh g^−1^	[94]
PET/Hollow silica	PVDF-HFP	Dip coating	22/1 M LiPF_6_^−^ EC-DEC (1:1, V:V)	2.57 mS cm^−1^/150 °C/0%	LiFePO_4_/Li	**In 2.5–4.2 V** 0.5 C ≈ 141 mAhg^−1^ 1 C ≈ 137 mAhg^−1^ 2 C ≈ 127 mAhg^−1^ 3 C ≈ 112 mAhg^−1^ 8 C ≈ 90 mAhg^−1^	[95]
PPS/SiO_2_	PVDF-HFP	Dip coating	114/1 M LiPF_6_^−^ EC-DEC (1:1, V:V)	1.02 mS cm^−1^/250 °C/0%	LiFePO_4_/Li	0.2 C ≈ 145 mAh g^−1^ 0.5 C ≈ 140 mAh g^−1^ 1 C ≈ 135 mAh g^−1^ 2 C ≈ 83 mAh g^−1^	[96]
BC/Al_2_O_3_	-	Dip coating	30/1 M LiPF_6_^−^ EC-DMC-DEC (1:1:1, V:V:V)	4.91 mS cm^−1^/150 °C/0%	LiFePO_4_/Li	**In 2.5–4.2 V** 0.2 C ≈ 160 mAh g^−1^ 0.5 C ≈ 140 mAh g^−1^ 1 C ≈ 130 mAh g^−1^	[97]
PAN/SiO_2_	TEGDA	Dip Coating	35/1.15 M LiPF_6_^−^ EC-EMC-DEC (3:5:2, V:V:V)	2.1 mS cm^−1^/200 °C/0%	LiNi_0.6_Co_0.6_Mn_0.2_O_2_/Graphite	**In 2.6–4.3 V** Initial discharge capacity 172.5 mAh g^−1^ Final discharge capacity 162.1 mA h g^−1^ After 100 cycles.	[98]
PVDF/LATP	-	Casting method	-/1 M LiPF_6_^−^ EC-EMC-DMC (1:1:1, V:V:V)	-/0.967 mS cm^−1^/-	LiFePO_4_Li	**In 2.5–4.2 V** Optimum ratio (LATP:PVDF 2:1) 0.1 C ≈ 163.5 mAh g^−1^ 0.2 C ≈ 155 mAh g^−1^ 0.5 C ≈ 150 mAh g^−1^ 1 C ≈ 141 mAh g^−1^	[105]
PVDF/SiO_2_	-	Phase inversion	-/1 M LiPF_6_^−^ EC-DMC (1:1, V:V)	-/0.9 mS cm^−1^/-	LiFePO_4_Li	**In 2.5–4.2 V** 0.1 C ≈ 150 mAh g^−1^ 0.2 C ≈ 149 mAh g^−1^ 0.5 C ≈ 140 mAh g^−1^ 10 C ≈ 128mAh g^−1^ 2 C ≈ 119mAh g^−1^	[106]
PSA/SiO_2_	-	Phase inversion	40/1 M LiPF_6_^−^ EC-DMC (1:1, V:V)	0.748 mS cm^−1^/150 °C/0%	LiCoO_2_/Li	0.2 C ≈ 146 mAh g^−1^ 0.5 C ≈ 132 mAh g^−1^ 2 C ≈ 113 mAh g^−1^ 5 C ≈ 95 mAh g^−1^ 10 C ≈ 70 mAh g^−1^	[107]
PVDF-HFP/Al_2_O_3_	-	Solvent evaporation method	40–45/1 M LiPF_6_^−^ EC-DMC-EMC-DMC +2% VC	0.7 mS cm^−1^/150 °C/4.5%	LiFePO_4_/Li	**In 2.5–4.2 V** 0.2 C ≈ 155 mAh g^−1^ 0.5 C ≈ 152 mAh g^−1^ 1 C ≈ 139 mAh g^−1^ 2 C ≈ 120 mAh g^−1^	[110]
P(VDF-TrFE) P(VDF-TrFE) 4% MMT 16% NaY 16% BaTiO_3_ 0.1%MWCNT	-	Solvent casting method	-/1 M LiTFSI PC	-/0.32 mS cm^−1^/- 0.36 mS cm^−1^ 0.39 mS cm^−1^ 0.64 mS cm^−1^ 0.33 mS cm^−1^	LiFePO_4_/Li	For 4% MMT 0.1 C-175 mAh g^−1^ 0.2 C ≈ 169 mAh g^−1^ 0.5 C ≈ 157 mAh g^−1^ 1 C ≈ 140 mAh g^−1^ 2 C ≈ 95 mAh g^−1^	[111]
PVDF/CA/Al(OH)_2_	-	Phase inversion	-/1 M LiPF_6_^−^ EC-DMC-EMC (1:1:1, W:W:W)	2.85 mS cm^−1^/160 °C/4.6 %	LiCoO_2_/Li	**In 3.0–4.2 V** 0.1 C ≈ 158 mAh g^−1^ 0.2 C ≈ 155 mAh g^−1^ 0.5 C ≈ 151 mAh g^−1^ 1 C ≈ 149 mAh g^−1^ 2 C ≈ 145 mAh g^−1^ 4 C ≈ 135 mAh g^−1^ 8 C ≈ 128.28 mAh g^−1^	[112]
SBR/Al_2_O_3_	-	Phase inversion method	37/1 M LiPF_6_^−^ EC-DMC (1:1, W:W)	0.93 mS cm^−1^/130 °C/0 %	LiNi_1/3_Co_1/3_Mn_1/3_O_2_/graphite	0.5 C ≈ 155 mAh g^−1^ 1 C ≈ 151 mAh g^−1^ 2 C ≈ 148 mAh g^−1^ 4 C ≈ 140 mAh g^−1^ 8 C ≈ 128 mAh g^−1^	[113]
PVDF/TiO_2_	-	Electrospinning	50/1 M LiPF_6_^−^ EC-DEC (1:1 V:V)	-/4.15 mS cm^−1^/-	-	High electrolyte uptake, high ion conductivity, wide electrochemical window	[117]
PC4SA-co-PMMA-co-PMPS/ZCM-5	-	Electrospinning	-/1 M LiPF_6_^−^ EC-EMC-DEC (1:1:1, V:V:V)	1.72 mS cm^−1^/150 °C/0%	LiFePO_4_/Li	**In 2.5–4.2 V** 0.1 C ≈ 153 mAh g^−1^ 0.2 C ≈ 146 mAh g^−1^ 0.5 C ≈ 140 mAh g^−1^ 1 C ≈ 130 mAh g^−1^ 2 C ≈ 120 mAh g^−1^ 5 C ≈ 110 mAh g^−1^	[118]
PAN/SiO_2_	-	Electrospinning	-/1 M LiPF_6_^−^ EC-DMC-DEC	-/1.04 ± 0.05 mS cm^−1^/-	LiCoO_2_/Graphite	**In 2.5–4.2 V** 0.1 C ≈ 135 mAh g^−1^ 0.2 C ≈ 130 mAh g^−1^ 0.5 C ≈ 126 mAh g^−1^ 1 C ≈ 123 mAh g^−1^ 2 C ≈ 115 mAh g^−1^ 5 C ≈ 72 mAh g^−1^	[119]
PVDF/MMT	-	Electrospinning	-/1 M LiPF_6_^−^ EC-EMC-DEC (1:1:1, V:V:V)	4.20 mS cm^−1^/150 °C/13.5%	LiFePO_4_/Li	**In 2.5–4.2 V** 0.2 C ≈ 157 mAh g^−1^ 0.5 C ≈ 139 mAh g^−1^ 1 C ≈ 129 mAh g^−1^ 2 C ≈ 115 mAh g^−1^	[120]
Cellulose/HAP	-	Electrospinning	56/1 M LiPF_6_^−^ EC-EMC (1:1:1, V:V:V)	-/200 °C/0%	LiFePO_4_/Li	0.5 C ≈ 144 mAh g^−1^ 1 C ≈ 139 mAh g^−1^ 2 C ≈ 130 mAh g^−1^ 3 C ≈ 125 mAh g^−1^ 4 C ≈ 120 mAh g^−1^ 5 C ≈ 118 mAh g^−1^	[121]
PMIA/Octaphenyl-POSS	-	Electrospinning	90–110 μm/1 M LiPF_6_^−^ EC-DMC (1:1 V:V)	-/1.93 mS cm^−1^/240 °C/0%	LiCoO_2_/Li	0.1 C ≈ 167 mAh g^−1^ 0.5 C ≈ 159 mAh g^−1^ 1 C ≈ 150 mAh g^−1^ 1.5 C ≈ 137.5 mAh g^−1^ 2 C≈119 mAh g^−1^	[122]
PI/SiO_2_	-	Electrospinning	20/1 M LiPF_6_^−^ EC-DEC (1:1, V:V)	2.27 mS cm^−1^/250 °C/0%	LiMn_2_O_4_/Li	**In 3.5–4.3 V** 0.2 C ≈ 110 mAh g^−1^ 0.5 C ≈ 107 mAh g^−1^ 1 C ≈ 103 mAh g^−1^ 2 C ≈ 99 mAh g^−1^ 5 C ≈ 82 mAh g^−1^ 10 C ≈ 55 mAh g^−1^	[123]
SiO_2_@PI/m-PE/SiO_2_@PI	-	Electrospinning + Dip coating	32/1 M LiPF_6_^−^ EC-DEC ( 1∶1, V:V)	0.941 mS cm^−1^/-/-	LiCoO_2_/Li	**In 2.8–4.2 V** Initial discharge capacity: 162.4 mAh g^−1^ Capacity retention: 83.5% after 100 cycles at 0.2 C	[125]
PVDF-HFP/Al_2_O_3_	-	Electrospinning +Atomic layer deposition	42 ± 2/ 1 M LiClO_4_ (EC-DMC, W:W)	1.24 mS cm^−1^/270 °C/0%	LiMn_2_O_4_/graphite	**In 3–4.2 V** 0.5 C ≈ 199 mAhg^−1^ 1 C ≈ 199 mAhg^−1^ 2 C ≈ 196 mAh g^−1^ 3 C ≈ 192 mAh g^−1^ 10 C ≈ 186 mAh g^−1^	[126]
PVP/TiO_2_	-	Electrospinning	-/1 M LiPF_6_^−^ EC-DEC-DMC (1:1:1, V:V:V)	1.27 mS cm^−1^/500 °C/0%	LiFePO_4_/Li	**In 2.5–4.2 V** 0.2 C ≈ 143 mAh g^−1^ 0.5 C ≈ 120 mAh g^−1^ 1 C ≈ 108mAh g^−1^ 2 C ≈ 90 mAh g^−1^ 3 C ≈ 70 mAh g^−1^ 5 C ≈ 50 mAh g^−1^	[127]
PAN/SiO_2_	-	Electrospinning	65/1 M LiPF_6_^−^ EC-EMC (1:1, V:V)	2.6 mS cm^−1^/150 °C/0%	LiFePO_4_/Li	**In 2.5–4.2 V** 0.2 C ≈ 163 mAh g^−1^ 0.5 C ≈ 157 mAh g^−1^ 1 C ≈ 142 mAh g^−1^ 2 C ≈ 127 mAh g^−1^ 4 C ≈ 118 mAh g^−1^ 8 C ≈ 83 mAh g^−1^	[128]

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
