# Peer review of "A Review on Inorganic Nanoparticles Modified Composite Membranes for Lithium-Ion Batteries: Recent Progress and Prospects"

_membranes, 2019, doi:10.3390/membranes9070078_

Round 1

Reviewer 1 Report

The authors reviewed the recent progress and prospects of inorganic nanoparticles modified composite membranes for lithium-ion batteries. The manuscript can be published in Membranes.

Line 16, fix “inorganic coated membrane, inorganic-filled membrane and inorganic-filled non-woven mates” as “inorganic particle-coated composite membranes, inorganic particle-filled composite membranes and inorganic particle-filled non-woven mates” (as the titles of 2, 3 and 4 sections)

Line 26, fix “Li-ion mechanism” as Li-ion battery mechanism”.

Line 38, fix “LIBs” as “lithium ion batteries” or use the “LIB” for the whole paper.

Line 49, fix “glass transition temperature” as “melting temperature”.

Line 74, please describe the three categories as in Abstract.

Line 169, ALOOH should be AlOOH.

Figures 2-8 are too small to read.

Author Response

We appreciate the reviewer’s comments and suggestions. We have been rectified all reviewer suggested mistakes in the modified manuscript. The modified text can be seen by” track changes” function in the MS word file.

Reviewer 2 Report

The subject is very interesting and addresses a real need for synthesis and comparisons between the different current techniques on inorganic nanoparticles modified composite membranes for Li-ion batteries.
The review is quite exhaustive and will provide a good basis for reflection for many researchers.
The division of section 2 will have to be reviewed: there is only one subtitle (2.1.). In my opinion, it will be necessary to have clearly identified paragraphs, one per type of technique.
The conclusion in Section 2 needs to be more thorough and critical.
Several lines of text are truncated. The layout will have to be reviewed in places.

Author Response

We appreciate the reviewer comments and suggestions. We have been modified all changes according to the reviewer suggestion in the modified manuscript.  The modified text, layout design and additional changes can be seen with “track changes” function in MS word.
